# A megawatt ultra-wide bandgap semiconductor module for pulsed power electronics

Hehe Gong[1,5], Xin Yang [1,5], Boyan Wang[2], Zichen Zhang[3], Qingrui Yuchi[3], Zineng Yang[1], Matthew Porter[2], Hongchang Cui[1], Yuan Qin[2], Rong Zhang [4], Han Wang [1], Dong Dong[2], Jiandong Ye [4]✉, Guo-Quan Lu[3]✉ & Yuhao Zhang [1]✉

Ultra-wide bandgap semiconductors exhibit advantageous electronic properties that make them promising for high-voltage, high-power electronics applications. Building on over a decade of progress in material growth and device fabrication, discrete ultra-wide bandgap devices with power-switching capacities up to the kilowatt level have been recently demonstrated. However, a packaged, multi-die ultra-wide bandgap power module – essential for further power scaling toward industrial, biomedical, grid, and aerospace applications – has yet to be realized. Here, we present a flip-chip packaged gallium oxide power module capable of 1000 A, 1000 V pulsed power switching with fast speed and minimal reverse recovery, advancing the power capacity of ultra-wide bandgap electronics by over two orders of magnitude. To address challenges posed by high electric fields and transient power surges, we employ a high-permittivity interface design enabling device-package electrothermal co-optimization. This optimization maximizes the module's transient thermal performance and enables full exploitation of the high volumetric heat capacity of gallium oxide—a largely untapped advantage in prior device development—alongside its high-temperature stability. The optimized ultra-wide bandgap module achieves over 1.8 MW/cm$^2$ pulsed power capacity density, outperforming silicon and wide-bandgap semiconductor counterparts and suggesting the promise of ultra-wide bandgap electronics in next-generation high-power systems.

Power semiconductor devices, which conduct high current in the on-state, block high voltage in the off-state, and fast switch between these two states, are central components in power electronics systems in numerous applications such as consumer electronics, data centers, electric vehicles, renewable energy processing, and electric grids. The global market of power semiconductor devices and modules has exceeded US\$50 billion[1]. In the last two decades, wide bandgap (WBG) semiconductor devices based on gallium nitride (GaN) and silicon carbide (SiC) have enabled system performance beyond the limits of conventional silicon (Si)-based devices. Recently, ultra-wide bandgap

[1]Centre for Advanced Semiconductors and Integrated Circuits and Department of Electrical and Computer Engineering, The University of Hong Kong, Hong Kong SAR, China. [2]Center for Power Electronics Systems, Virginia Polytechnic Institute and State University, Blacksburg, VA, USA. [3]Department of Materials Science and Engineering, Virginia Polytechnic Institute and State University, Blacksburg, VA, USA. [4]School of Electronic Science and Engineering, Nanjing University, Nanjing, China. [5]These authors contributed equally: Hehe Gong, Xin Yang. ✉e-mail: yejd@nju.edu.cn; gqlu@vt.edu; yuhzhang@hku.hk

(UWBG) semiconductors, including gallium oxide ($Ga_2O_3$), aluminum nitride (AlN), and diamond, have emerged as competitive candidates for next-generation power devices. These materials promise higher critical electric field and thermal stability compared to the WBG and Si counterparts[2,3]. Their Baliga's figure of merit (FOM)[4], the FOM widely used to evaluate semiconductor materials for power applications, are at least 3.8 times higher than GaN or SiC and 3000 times than Si[5].

Over the past decade, the development of UWBG power devices has advanced rapidly in power scaling, driven by the availability of large-diameter wafers and progress in device fabrication techniques[6–10]. For example, 8-in. $Ga_2O_3$ wafer and $32 \times 31.5$ mm² diamond wafer have been recently demonstrated[11,12]. Fabricated UWBG devices have exhibited promising switching performance in circuit-level evaluations, with switching power capacity ($P_C$, defined as the product of switching voltage and current) reaching the kilowatt range. For instance, $Ga_2O_3$ diodes have been tested in a buck converter delivering output power up to 2 kW, and in double pulse tests with switching voltage and current of 400 V and 15 A, respectively[13]. Additionally, diamond diodes have demonstrated short reverse recovery times during transitions from 2 A forward current to 100 V blocking voltage[14].

However, the current power levels achieved by UWBG devices remain insufficient for the demands of industrial, biomedical, grid, and aerospace applications. For example, next-generation electric vehicle platforms support charging voltages up to 1000 V and currents up to 1000 A, requesting a megawatt (MW) $P_C$ for power devices[15]. In certain industrial and aerospace applications, such as electromagnetic forming[16] and electromagnetic propulsion[17], transient power levels can exceed 100 MW, with peak currents surpassing 100 kA. Traditionally, high power in Si devices is achieved by processing an entire wafer into a single, large-area device. However, this approach is not viable for WBG and UWBG materials due to intrinsic material non-uniformities, fabrication variability, and high defect densities[18]. These limitations often introduce a trade-off between breakdown voltage ($V_B$) and forward current ($I_F$). For example, in large-area $Ga_2O_3$ devices, $V_B$ typically decreases as $I_F$ increases[6] (Supplementary Section S1), constraining power scaling.

As an alternative, power modules that integrate multiple devices with advanced packaging and thermal management are commonly adopted in the WBG industry[19,20]. However, no such power modules have yet been demonstrated for UWBG devices, primarily due to several key challenges: (1) packaging for UWBG devices must withstand very high electric fields near device edges and encapsulation interfaces; and (2) these devices must support very high power densities while maintaining a compact footprint, resulting in high heat fluxes that are difficult to manage[21]. These challenges are compounded by the inherently low thermal conductivity ($k_T$) of many UWBG materials; for instance, $Ga_2O_3$ has a low $k_T$ of 11–27 W/m K[22]. As a result, the power level of UWBG devices has stagnated at the kilowatt scale for nearly a decade. Additionally, the absence of a demonstrated high-power module suitable for industrial deployment continues to limit the practical application space of UWBG devices. Notably, although UWBG materials can support high-temperature operation due to their low intrinsic carrier concentrations[9], this advantage remains largely unexploited because conventional packaging designs are often inadequate for such thermal environments.

This work addresses the critical limitation in power scaling of UWBG devices by introducing a $Ga_2O_3$ power module through device-package, electrothermal co-design. Building on our earlier conference report[23], a high-permittivity ($\kappa$) $BaTiO_3$ layer was employed as an interface between the device's edge termination and the die attach of the package, inducing polarization-induced dipoles that effectively flatten the localized electric field crowding. This design further eliminates the need for the posts that are typically used in conventional flip-chip packages, improving thermal resistance by over 50%. The

packaged $Ga_2O_3$ diode exhibits a high $V_B$ over 2 kV, high current capability over 100 A, low thermal resistance of 0.47 K/W, and stable operation up to 250 °C—collectively representing state-of-the-art performance among UWBG power devices reported to date.

Furthermore, the co-design enables exploration of emerging application avenues for UWBG devices—pulsed power electronics. These applications—spanning defense, biomedicine, industrial processing, scientific instrumentation, energy systems, and aerospace—rely on the generation and control of short, repetitive, high-power electrical pulses. We found that $Ga_2O_3$ power modules outperform their Si and WBG counterparts in pulse power scenarios, owing to three key factors: (a) $Ga_2O_3$'s high volumetric heat capacity ($C_v$), a critical property largely overlooked in prior studies; (b) the intrinsic capability of $Ga_2O_3$ device to operate at elevated temperatures; and (c) the co-designed package structure, which is optimized for transient thermal management. We demonstrated continuous pulsed switching at 1000 V and 1000 A—a $P_C$ level that exceeds previously reported UWBG devices by more than two orders of magnitude.

## Results

### Pulsed power applications and material selection

The global market size of pulsed power applications is projected to reach US$100 billion by 2032[24]. A typical pulsed power system consists of two core stages—low power accumulation and high power output—through a pulse forming network, which delivers megawatt- to gigawatt- pulses over very short period[25] (Fig. 1a). Such high-power pulse supplies are essential in diverse applications, including grid, magnetic resonance imaging (MRI), pulsed lasers, and nuclear fusion systems. For example, in grid-connected circuit breakers, devices are required to withstand a concurrence of high voltage and surge current on the microsecond timescale before the mechanism branch intervenes[26,27]. Compared to conventional power switching systems, pulsed power systems impose significantly higher transient power demands, making transient thermal performance far more critical than steady-state behavior[28].

The performance of pulsed power systems is critically determined by the switching device connecting energy storage and pulse delivery stages, which must handle large transient currents, high voltages, fast switching speeds, and elevated transient temperatures[29]. Currently, pulsed power devices fall into two main categories: gas discharge devices and solid-state devices (Fig. 1b)[30]. Despite high power handling capabilities, gas discharge devices suffer from low repetition rates, limited lifetimes, and large physical footprints. In solid-state devices, Si bipolar devices can handle high power but exhibit slow switching speeds, high conduction loss, and limited high-temperature stability. Emerging SiC unipolar devices offer faster switching speeds, but their power handling remains constrained. Next-generation pulsed power devices, including both controlled switches (i.e., power transistors) and power diodes, are therefore in high demand. In pulsed-power systems, transistors are often used for initiating and shaping the pulse, whereas power diodes play indispensable roles in energy steering, stage isolation, rapid charge management, and transient overvoltage suppression. For instance, in inductive-load pulsed circuits, the diode must withstand and conduct large surge currents while clamping voltage spikes induced by rapid current transients ($L \cdot di/dt$), all within several microseconds. In high-voltage pulse generation circuits, such as Marx and Cockcroft-Walton generators, the diode is further required to reliably block kilovolt reverse voltages.

Semiconductor material selection for pulsed power devices must account for both electrical FOMs and thermal properties. For thermal considerations, during short-duration pulsed heating, the heat has insufficient time to spread spatially, and thus the temperature rise is governed primarily by the material's $C_v$ instead of $k_T$. As thermal diffusion is negligible, an approximate model dictates the local temperature rise $\Delta T$ to scale with $Q/(C_v V)$[31], where $Q$ is the heat generation

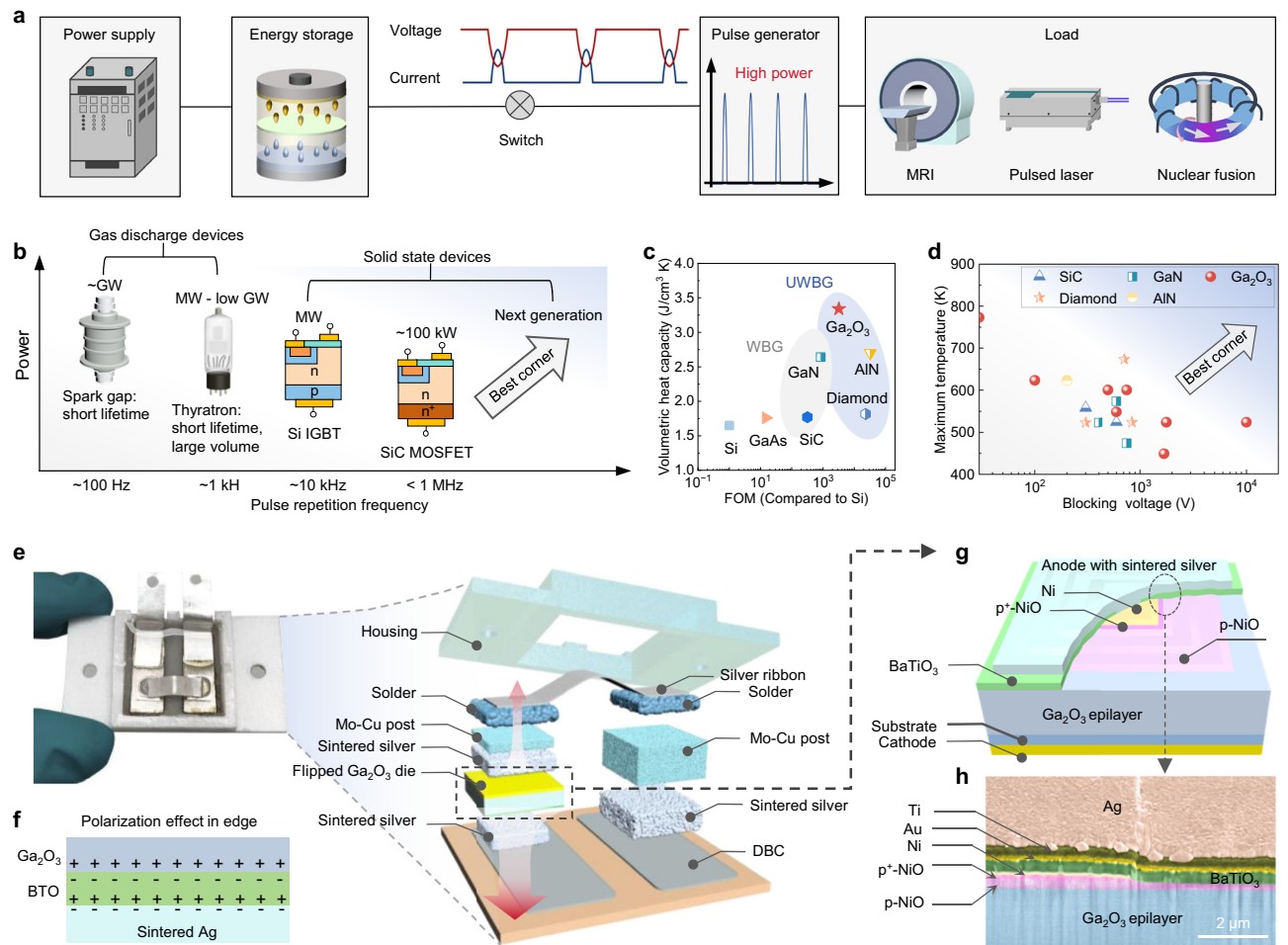

**Fig. 1 | Pulsed power applications, material selection, as well as device and package designs. a** Schematic of a typical pulsed power system for diverse applications and the illustration of operating voltage and current for the switch. **b** The power level and pulse repetition frequency of current pulsed power devices, as well as illustration of the desirable characteristics of the next-generation devices. **c** Comparison of the volumetric heat capacity and Baliga's FOM for mainstream power semiconductors at room temperature, including Si, GaAs, SiC, GaN, $Ga_2O_3$, AlN, and diamond. Data for volumetric heat capacity of typical power semiconductors are from refs. 34–41. **d** Benchmark of the reported maximum operational temperature versus the maximum blocking voltage at this temperature for the state-the-art WBG and UWBG power devices. Data are from refs. 43–59. **e** Photograph and three-dimensional schematic of the fabricated package architectures in a $Ga_2O_3$ sub-module. **f** Illustration of the polarization dipoles introduced by the high-$\kappa$ dielectrics in the metal/$BaTiO_3$/$Ga_2O_3$ structure. **g, h** Three-dimensional schematic and cross-sectional SEM images (in false color) of the high-$\kappa$ interface region, where a $BaTiO_3$ layer is sandwiched between the NiO JTE and the overflowed sintered silver.

energy, and $V$ is the material volume. Hence, a higher $C_v$ indicates a smaller temperature increase under the identical power loss[32], offering improved thermal buffering against fast surges[33]. Figure 1c compares the room-temperature $C_v$ and Baliga's FOM of representative power semiconductors, including Si[34], GaAs[35], SiC[36], GaN[37], $Ga_2O_3$[38,39], AlN[40], and diamond[41]. $Ga_2O_3$ exhibits the largest $C_v$ – nearly twice that of SiC – and one of the highest Baliga's FOMs, making it optimal for pulsed switches.

Another fundamental limitation of transient power operation is the device's high-temperature operation capability, which is usually constrained by its high-temperature, high-bias blocking capability. Elevated temperatures enhance intrinsic carrier generation and activate traps, resulting in increased leakage current, localized self-heating and premature breakdown. As the intrinsic carrier density ($n_i$) increases exponentially with decreasing material bandgap, lower leakage current is expected in UWBG devices, enabling them to sustain higher operational temperatures[42]. Figure 1d compares the maximum operational temperature and blocking voltage reported for bare-die WBG[43–47] and UWBG power devices[48–59]. $Ga_2O_3$ devices exhibit the favorable combinations of high voltage and high temperature

performance, further consolidating their suitability for pulsed power applications.

## Device-package electrothermal co-optimization

Figure 1e shows the design of the packaging architecture for a single-die $Ga_2O_3$ device, which is referred to as a sub-module in this work, as well as the photography of a prototyped package. As the p-n junction enables lower leakage current and higher operational temperatures than metal-semiconductor junctions, a p-NiO/n-$Ga_2O_3$ p-n heterojunction diode (HJD) is selected for module development. Notably, this p-NiO/n-$Ga_2O_3$ p-n junction is also a common building block of various high-temperature, high-voltage $Ga_2O_3$ devices benchmarked in Fig. 1d[48–55]. To extract heat directly from the device junction instead of through the low-$k_T$ $Ga_2O_3$ chip, the junction-side cooling (JSC) package is employed, and the flip-chip $Ga_2O_3$ die is directly attached to the direct bond copper (DBC) plate. A molybdenum-copper (Mo-Cu) alloy post with a height similar to the $Ga_2O_3$ chip thickness is placed on the side and connected to the cathode of $Ga_2O_3$ HJD via an Ag ribbon. Benefitting from the smaller coefficient of thermal expansion (CTE) of the Mo-Cu alloy and the bend of the ribbon structure[60], these designs

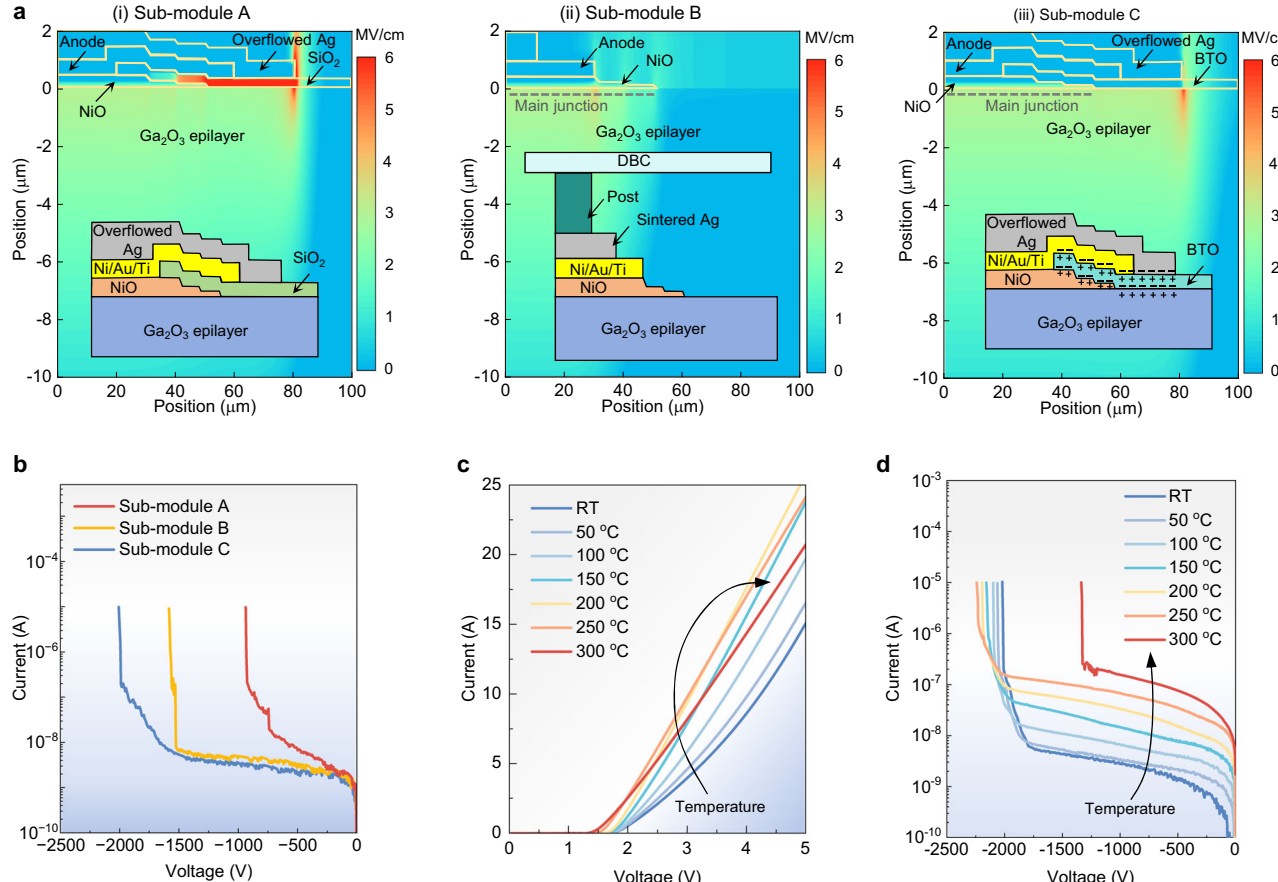

**Fig. 2 | Breakdown voltage enhancement and sub-modules' electrical characteristics. a** Simulated two-dimensional electric field contours in JSC-packaged NiO/Ga$_2$O$_3$ heterojunction devices (i) with a SiO$_2$ interface (control sub-module A), (ii) with a post interface (control sub-module B), and (iii) with a high-$\kappa$ interface (proposed design, referred to as sub-module C), all simulated under a reverse bias of 2000 V. The inset of each sub-figure illustrates the schematic of the package structure near the edge termination region. For clarity, the flip-chip packaged devices are rotated by 180° to display the edge region more clearly. **b** Reverse I-V characteristics of three types of Ga$_2$O$_3$ sub-modules with different package structures. **c**, **d** Temperature-dependent forward and reverse I-V characteristics of the Ga$_2$O$_3$ sub-module with high-$\kappa$ interface.

can absorb the thermo-mechanical stress induced by cyclic temperature variations, and thereby boost the reliability and lifetime of the sub-module. Ag sintering is deployed as the primary die attach, with solder only used for connecting the Ag ribbon.

While the JSC package has been demonstrated as an effective thermal management approach for Ga$_2$O$_3$ devices[61,62], a widely-reported adverse bi-product is a degradation in device $V_B$[63]. This is usually induced by the inevitable overflow of the die attach materials—such as soldering and sintering metals—from device active region to the edge area, producing additional and higher electric field peaks in device[61]. The optical image of overflowed sintered Ag after JSC package is shown in Supplementary Section S2. The conventional solution to this problem is adding a post with sufficient height to physically distance the attach materials from the device surface. However, the added post can introduce substantial thermal resistance, compromising the effectiveness of the JSC package in heat extraction.

To address this dilemma, we replace the post with a high-$\kappa$ interface inserted between the device edge termination and the overflown die attach material, which could concurrently improve the device's electrical and thermal performance. As illustrated in Fig. 1f, owing to the large permittivity contrast between BTO and Ga$_2$O$_3$, the external electric field induces a polarization gradient inside the BTO layer, creating net negative bound polarization charge at the BTO/Ga$_2$O$_3$ interface and a corresponding positive charge beneath the metal/BTO interface[64,65]. This polarization dipole can shield the electric

field produced by charges in the metal and enable a nearly-zero net charge to flatten the crowded electric field—similar to the legacy superjunction design in multidimensional power devices[66]. Figure 1g illustrates the prototype high-$\kappa$ interface for Ga$_2$O$_3$ HJD, and Fig. 1h shows the cross-sectional scanning electron microscopy (SEM) image. The device's edge termination consists of multiple lightly-doped p-NiO layers with varied lengths, constructing a graded junction termination extension (JTE) that enables a graded decrease in charge density away from the main junction to spatially spread the electric field[67]. On top of this NiO JTE and below the overflowed Ag, a 300 nm-thick BaTiO$_3$ layer with $\kappa$ of over 155 (Supplementary Section S3) is deposited through magnetron sputtering. The detailed processes for device fabrication and packaging are illustrated in "Methods" section and Supplementary Sections S4 and S5.

To validate the functionality of the high-$\kappa$ interface design, we fabricated two additional packages as control samples, one with the conventional SiO$_2$ passivation layer (control sub-module A) and the other with an additional 500 μm-thick Mo-Cu post (control sub-module B); the high-$\kappa$ interface submodule is referred to as sub-module C. Physics-based TCAD simulation is performed to obtain the electrical field contours of three packages at a blocking voltage of 2000 V. The physical parameters and models in simulation are detailed in Supplementary Section S6. Simulation reveals a peak electric field up to 19 MV/cm in SiO$_2$ in the control sub-module A, which exceeds the critical electric field of SiO$_2$ (Fig. 2ai). Such an electric field peak can be

effectively suppressed by either a post in the control sub-module B (Fig. 2aii) or the proposed high-$\kappa$ interface (Fig. 2aiii). Figure 2b shows the experimental reverse current-voltage (I-V) characteristics of the three packaged $Ga_2O_3$ HJDs, revealing a $V_B$ of 930 V, 1560 V and 2010 V in the sub-modules A−C. The $V_B$ enhancement verifies the effectiveness of high-$\kappa$ interface for electric field management.

Figure 2c, d shows the temperature-dependent forward and reverse I-V characteristics of the sub-module C, respectively. The $Ga_2O_3$ sub-module exhibits an enhanced current capability due to high-level conductivity modulation as temperature ($T$) increases from room temperature (RT) to 200 °C[68], while the differential on-resistance ($R_{ON}$) gradually increases from 200 to 300 °C, attributed to the dominance of mobility degradation over conductivity modulation. The $V_B$ increases from 2010 to 2240 V with $T$ from RT to 250 °C[48], exhibiting a positive temperature coefficient of ~1 V/°C, which suggests the avalanche capability. This operation temperature is among the highest reported for kilovolt, large-area UWBG devices. When the temperature is further increased to 300 °C, $V_B$ decreases to 1330 V, which is likely due to the trap-assisted preliminary breakdown possibly originating from the interface states between Ag, BTO, and $Ga_2O_3$. Such a trap-assisted breakdown has been widely reported in other ultra-wide bandgap power devices and features a negative temperature coefficient[69,70]. As a comparison, the temperature-dependent reverse I-V characteristics of a commercial SiC power diode show higher leakage current and significantly degraded $V_B$ when $T$ is above 150 °C — for example, $V_B$ drops to 61% of the RT value at 200 °C (Supplementary Section S7).

## Transient thermal impedance and maximum power capacity

Thermal impedance ($Z$) was measured to compare the junction-to-case thermal resistance ($R_{\theta JC}$) of the conventional $Ga_2O_3$ sub-module B with post and the proposed sub-module C with high-$\kappa$ interface. The measurement is based on the transient dual interface method (TDIM)—an industry standard (JEDEC 51-14) for characterizing the $R_{\theta JC}$ of packaged power devices[71]. This method relies on capturing transient thermal responses of the packaged devices using two thermal interface materials (TIMs). From the resulting time evolution of transient thermal impedance ($Z$ ~ $t$ curve), both transient thermal impedance and the steady-state thermal resistance can be extracted, with the latter determined from the separation point of the two $Z$ ~ $t$ curves. The experimental measurement setups are described in the "Methods" section and Supplementary Section S8. Figure 3a presents the measured $Z$ ~ $t$ curves of the two $Ga_2O_3$ sub-modules, each with two TIMs, revealing a steady state $R_{\theta JC}$ of 1.05 K/W and 0.47 K/W for sub-modules with the post and the high-$\kappa$ interface, respectively. The comparison reveals that, in the JSC package, the elimination of post enabled by high-$\kappa$ interface allows for an $R_{\theta JC}$ reduction by over 50%.

Physics-based thermal simulations in ANSYS are performed to uncover the internal temperature contours within the packaged sub-module and their time-resolved evolutions. The simulations are calibrated against the experimental case temperature ($T_C$) of the $Ga_2O_3$ sub-module measured across three orders of magnitudes in time scales under continuous 10 A, 1000 V circuit operations. The simulated temperature evolution closely matches experimental data under two cooling schemes (natural convection and fan cooling). The simulation models and calibrations are detailed in the "Methods" section and Supplementary Section S9. Figure 3b shows the simulated steady-state temperature contours inside two $Ga_2O_3$ sub-modules under identical 20 W power dissipation and cooling conditions. Compared to the high-$\kappa$ interface module with a nearly uniform junction temperature ($T_j$) of 57 °C, the sub-module with post shows a higher $T_j$ of 74 °C and an additional peak temperature of 89 °C near the anode edge. These hot spots arise because the post-contact area is smaller than the actual anode area, resulting in ineffective heat extraction at the anode edge regions.

To probe the sub-module's transient thermal performance, we simulate temperature contours in the two sub-modules under three pulse durations (10 μs, 1 ms, and 100 ms) while maintaining a constant dissipated energy of 0.5 J. As shown in Fig. 3c, the high-$\kappa$ interface module effectively suppresses hot spots at the die edge and lowers peak temperatures under all pulse conditions. Additionally, both sub-modules show higher $T_j$ for shorter pulses, indicating greater thermal stress and suggesting the critical importance of the device's maximum operational temperature in pulsed power applications.

The above thermal and electrical data allow for deriving the sub-module's maximum $P_C$ density at different time scales. Based on experimental $Z$ ~ $t$ characteristics and the 250 °C maximum operational temperature of $Ga_2O_3$ sub-module determined by voltage blocking capability (see Fig. 2d), the maximum power loss at different transient time scales can be calculated. This loss, in conjunction with experimental temperature-dependent I-V (I-V-T) characteristics, allows for further derivation of maximum forward current and $P_C$ at various time scales. This analysis is illustrated in Fig. 3d and detailed in Supplementary Section S10. Based on the same approach, the maximum $P_C$ under different time scales is also derived for an industrial SiC diode (IDH02G120C5) and a Si diode (RF305BM6S) with the same differential on-resistance (which indicates similar current conduction capability) compared to the $Ga_2O_3$ HJD, using their datasheet I-V-T and $Z$ ~ $t$ characteristics and assuming a maximum operational $T_j$ of 175 °C for SiC device and 150 °C for Si device. Figure 3e, f shows the derived time-resolved junction-to-case thermal impedance and maximum $P_C$ density, respectively, for three packaged devices. The maximum $P_C$ density of $Ga_2O_3$ diode sub-module outperforms the Si and SiC counterparts with increased advantages at shorter pulses, which is attributable to the higher $C_v$ of $Ga_2O_3$, higher blocking temperature of $Ga_2O_3$ devices, and the electrothermal co-optimized package structure.

## Circuit demonstration of megawatt pulsed-power switching

The projected superiority of $Ga_2O_3$ sub-module is explored experimentally in practical pulsed-power switching circuits. We designed an on-board double-pulse test (DPT) circuit for high-power inductive switching with adjustable pulse width, which represents the typical power rectifier operation in various pulsed-power applications. The DPT circuit consists of power SiC MOSFETs (C3M0016120D), an inductor, and the diode under test (DUT). The inductor is first precharged to a target current value, acting as a current source. The SiC MOSFETs are used to control short-pulse, high-voltage, and high-current waveforms with an adjustable pulse width, these pulses are then delivered to the $Ga_2O_3$ DUT for evaluating its turn-on and turn-off characteristics. The schematic and photo of the test setup are shown in Fig. 4a, b, respectively, and circuit working principle is detailed in Supplementary Section S11. Under a bus bias of 1000 V, the $Ga_2O_3$ sub-module is capable of conducting a forward current of 234 A with a pulse width of 5 μs, as shown in Fig. 4c. Time-resolved junction temperature simulations reveal that $T_j$ peaks at 251 °C for 234 A and 297 °C for 257 A during pulsed switching (Fig. 4d). At forward current over 234 A, the high $T_j$ starts to degrade device blocking voltage (see Fig. 2d). When the blocking voltage degrades below the bus voltage, breakdown occurs.

To further upscale the $P_C$ to reach that of industrial applications, we assemble a multi-die full module based on a symmetric arrangement of six $Ga_2O_3$ sub-modules (Fig. 4e). The module demonstrates reliable dynamic switching at 1000 V/1000 A under a switching frequency of 1 kHz (Fig. 4f, g), along with fast switching speed and minimal reverse recovery, with the turn-off time being extracted to be 23 ns. The thermal imaging results of the full module under continuous pulsed-power switching are shown in Supplementary Section S12, revealing uniform current sharing among six sub-modules. The case temperature reached 92 °C on the top and 81 °C on the bottom of each sub-module at the steady state, confirming the effective thermal management. Figure 4h

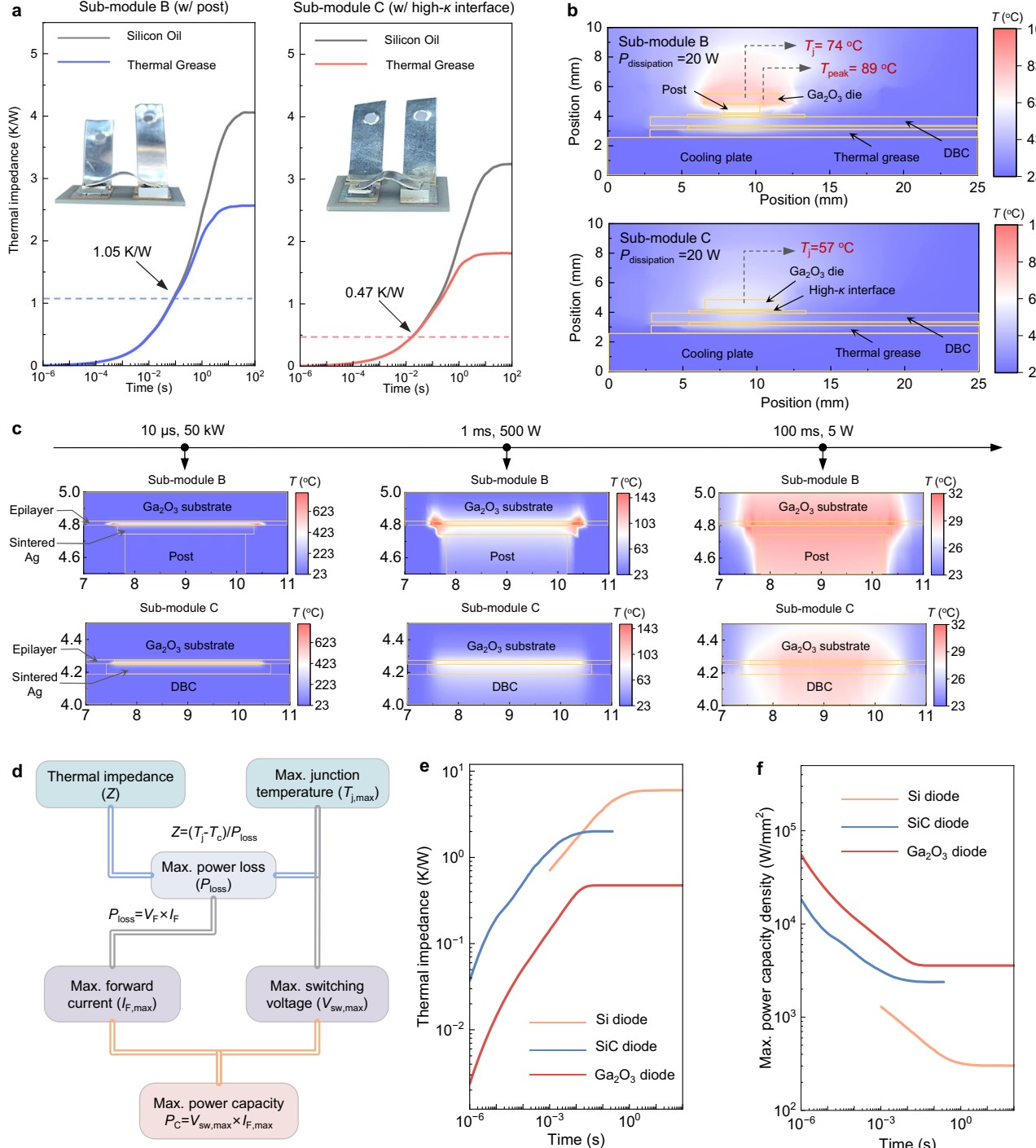

**Fig. 3 | Thermal impedance characterization and time-resolved maximum power capacity density derivation. a** Transient thermal impedance curves of sub-modules B and C; the insets show the optical images of the two prototyped sub-modules, where encapsulation is omitted to clearly reveal the package structure. **b** Simulated two-dimensional temperature contours under a steady state power dissipation ($P_{dissipation}$) of 20 W. **c** Transient temperature contours in the two sub-modules under an identical energy dissipation with three different pulse widths.

**d** Diagram illustrating the analysis of time-resolved maximum power capacity based on experimental Z - t curves, I-V-T characteristics, and the maximum $T_j$ determined by the high-bias blocking capability. **e** Derived transient thermal impedance of the $Ga_2O_3$ sub-module, packaged SiC diode (IDH02G120C5) and packaged Si diode (RF305BM6S), across various time scales. **f** Extracted maximum transient power capacity density for three packaged power diodes as across varying time scales from 1 μs to 100 s.

benchmarks the switching voltage and $P_C$ for the $Ga_2O_3$ sub-module and multi-die full module prototyped in this work, as well as the previously reported UWBG devices in practical circuit switching[13,14,57,72–75]. The $P_C$ of our $Ga_2O_3$ modules is over two orders of magnitude higher than the previous best report, validating the effectiveness of device-package co-optimization for UWBG modules.

## Short-pulse surge current capability and Power cycling test
From the analysis in Fig. 4d, the device failure in the kilovolt switching is limited by the maximum junction temperature for voltage blocking instead of the device's intrinsic thermal failure in forward current conduction. Meanwhile, in some pulse power applications (e.g., light laser detection and ranging (LiDAR) systems), devices are only

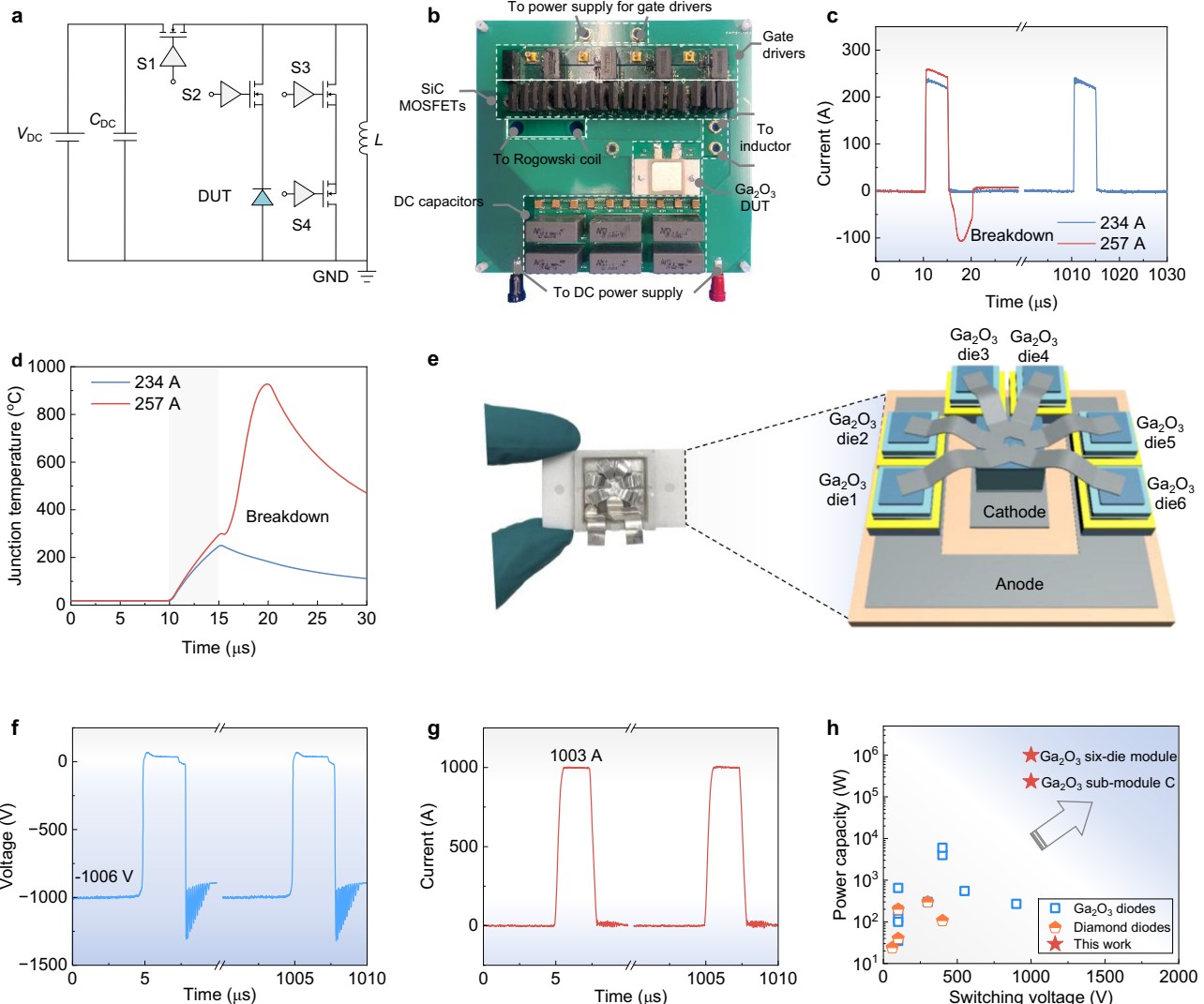

**Fig. 4 | Demonstration of megawatt pulsed power switching in practical power converter. a, b** Schematic and optical image of on-board pulsed power switching test circuit, which includes a DC power supply ($V_{DC}$), DC capacitance ($C_{DC}$), a load inductor ($L$), four power switches (S1-S4), and the Ga$_2$O$_3$ device under test (DUT). **c, d** Experimental switching current waveforms and simulated time-resolved $T_j$ evolution of the Ga$_2$O$_3$ sub-module with high-$\kappa$ interface. **e** Optical image and three-dimensional schematic of the prototyped full Ga$_2$O$_3$ module consisting of six sub-modules. **f, g** Switching voltage and switching current waveforms of the prototyped full Ga$_2$O$_3$ module showcasing the repetitive megawatt pulsed power switching. **h** Comparison of the maximum switching power capacity and switching voltage for our Ga$_2$O$_3$ modules and other UWBG devices reported in the literature. All data are extracted from reports of circuit tests and from refs. 13,14,57,72–75.

required to pass high current but do not need to block a very high voltage. For these applications scenarios, surge current capability is a key metric of power rectifiers. Here, we study the surge current capability of the Ga$_2$O$_3$ sub-module across a wide range of time spans to understand its intrinsic electrothermal limit in conduction-dominant pulsed power applications.

The schematic and photo of the surge-current evaluation circuit are shown in Fig. 5a, b, and the circuit working principle is detailed in Supplementary Section S13. Different from conventional surge current test circuits with a fixed pulse width (e.g., 10 ms), this test setup is tailored for pulse power applications by enabling adjustable pulse widths across four orders of magnitude down to 3.6 μs. The half-sinusoidal surge current signal is generated by adjusting the resonance circuit, and the pulse width is controlled by modulating the values of inductor and capacitor. The circuit layout is optimized to limit the diode's maximum ringing bias in the blocking state below 100 V.

Figure 5c shows the current and voltage waveforms in the surge current tests under a 3.6 μs pulse width and increased peak current amplitude. The Ga$_2$O$_3$ sub-module can withstand a maximum peak

surge current of 856 A. Post-test static characterization confirms the device integrity until the last 856 A surge current test, after which the device exhibits a measurable deterioration in its reverse-blocking capability. Figure 5d compares the surge current capabilities of the Ga$_2$O$_3$ sub-module under varying pulse widths. The critical peak surge current decreases from 856 to 65 A when the pulse width increases from 3.6 μs to 10 ms. The peak critical surge currents at various pulse widths are plotted in Fig. 5e from surge current tests performed under both the room temperature and an ambient temperature of 150 °C. Under two temperatures, the Ga$_2$O$_3$ sub-module delivers increasing current capacity for shorter pulse applications, and its current capacity consistently outperforms a state-of-the-art industrial 1200 V-rated SiC diode with a similar differential on-resistance (IDH02G120C5), particularly under shorter pulses and elevated temperatures.

Further physical insight is obtained via transient thermal simulations. Figure 5f, g present the surge power extracted from experimental waveforms, and the simulated evolutions of transient $T_j$ under critical surge current tests with varying pulse widths, respectively. Under the critical failure condition, the peak $T_j$ is projected to decrease

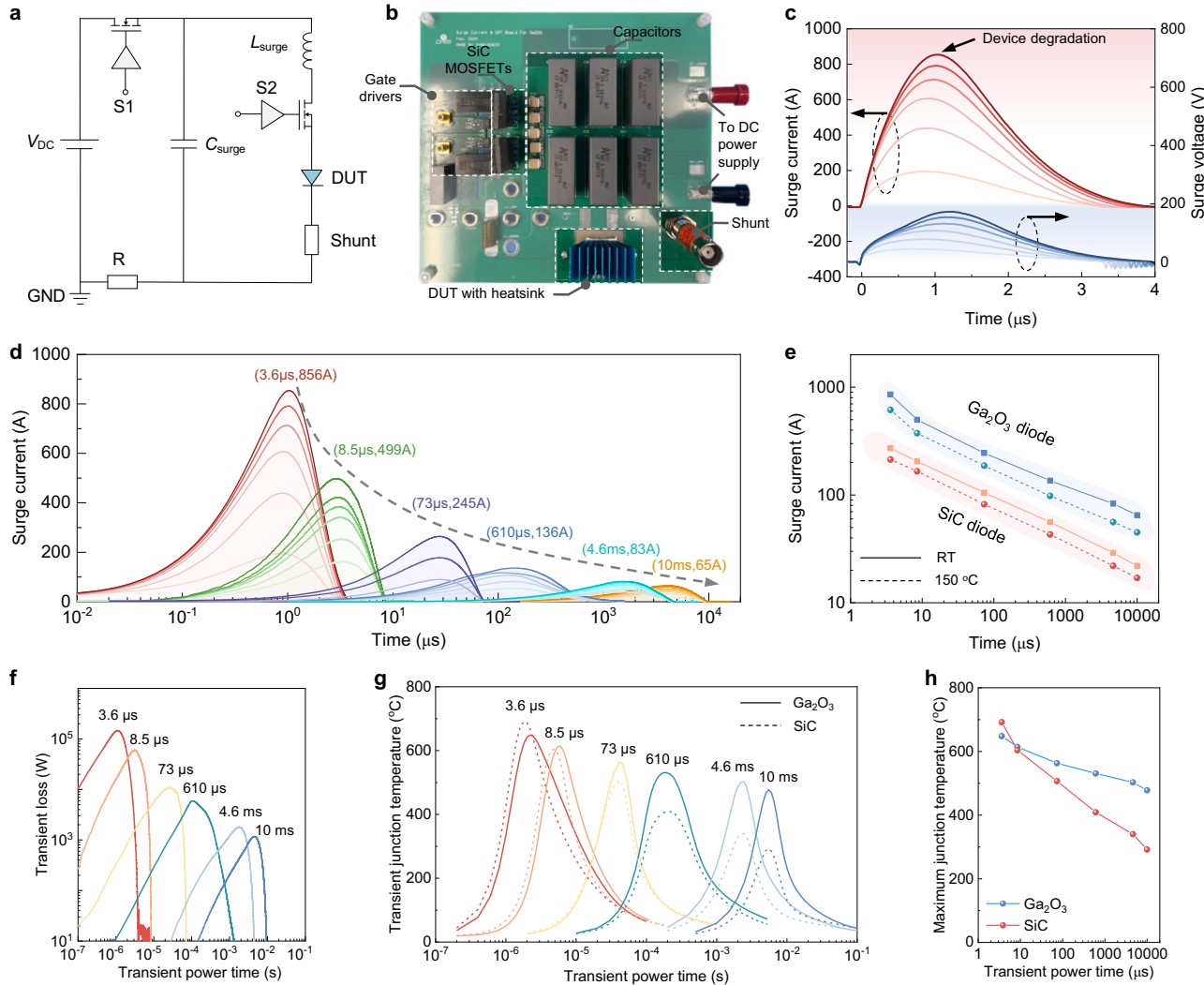

**Fig. 5 | Surge current capability of Ga$_2$O$_3$ sub-module in conduction-dominant pulsed power applications. a, b** Schematic and optical image of the surge current test circuit tailored for pulsed power operations, which includes a DC power supply ($V_{DC}$), the pulse-forming inductor ($L_{surge}$) and capacitor ($C_{surge}$), power switches, and DUT. **c** Surge current and surge voltage waveforms of the Ga$_2$O$_3$ sub-module tested with an increased surge energy at a fixed pulse width of 3.6 µs. The critical peak surge current is over 800 A. **d** Groups of surge current waveforms under increased surge energies tested at various pulse widths from 3.6 µs to 10 ms. The critical peak surge current under each pulse width is also marked. **e** Critical peak surge currents of Ga$_2$O$_3$ sub-module and a state-of-the-art industrial SiC diode (IDH02G120C5) measured as a function of pulse widths, all tested under two

environmental temperatures of room temperature and 150 °C. **f** Extracted transient power loss in surge current tests based on experimental voltage and current waveforms. **g** Simulated transient $T_j$ evolutions during critical surge current tests under various pulse widths for the Ga$_2$O$_3$ sub-module and a hypothetical SiC sub-module. The hypothetical SiC module has the same device and package structures compared to the Ga$_2$O$_3$ sub-module, but replaces the Ga$_2$O$_3$ material properties with the SiC's. **h** Extracted critical $T_j$ as a function of surge current pulse width for the Ga$_2$O$_3$ sub-module and the hypothetical SiC sub-module. The Ga$_2$O$_3$ sub-module can achieve a lower $T_j$ under short surge current pulses, leveraging the Ga$_2$O$_3$'s high volumetric heat capacity.

from 648 to 478 °C with pulse width increased from 3.6 µs to 10 ms, indicating that the Ga$_2$O$_3$ sub-module can sustain higher $T_j$ at short pulses. To further understand the superior performance of the Ga$_2$O$_3$ sub-module compared to its SiC counterpart, we also conduct transient thermal simulations on a hypothetical SiC sub-module that shares identical device and package architectures, with only the material properties substituted from Ga$_2$O$_3$ to SiC. The critical $T_j$ of Ga$_2$O$_3$ and SiC sub-modules under various pulse widths are summarized in Fig. 5h, and the evolution of internal temperature contours of two sub-modules in a critical surge current test is presented in Supplementary Section S14. At long pulse width, the hypothetical SiC sub-module shows lower critical $T_j$ than the Ga$_2$O$_3$ counterpart, primarily due to SiC's higher $k_T$. However, at short pulse width, the critical $T_j$ of the Ga$_2$O$_3$ sub-module falls below that of the SiC sub-module, highlighting the dominant influence of Ga$_2$O$_3$'s higher $C_v$ over its comparatively

lower $k_T$. This comparison further manifests the greater importance of a material's $C_v$—as well as the thermal capacitance of the package – in determining transient thermal behavior during pulsed power operations.

A similar consideration is also relevant for repetitive switching under varying duty cycles, or equivalently, different repetition rates. As shown in the analysis in Supplementary Section S15 at relatively high duty cycles or repetition rates, an average junction temperature rise develops in addition to the transient pulsed temperature peaks when the generated heat cannot be fully removed between pulses. This average junction temperature rise can be suppressed through improved external cooling. In this scenario, advanced external cooling is required to fully exploit the module's intrinsic power capacity.

Finally, we evaluate the reliability of the prototyped sub-modules. The BaTiO$_3$ layer is fully encapsulated by silicone gel, forming a

continuous dielectric barrier that suppresses humidity-induced degradation commonly observed in bulk $BaTiO_3$ ceramics[76]. In addition, the $BaTiO_3$ interface is found to improve thermo-mechanical reliability. Table S3 in Supplementary Section S16 summarizes the CTE of relevant materials. Due to the large CTE mismatch, conventional $Ga_2O_3/SiO_2$/Sintered Ag stacks experience elevated thermo-mechanical stress near die edges. Replacing $SiO_2$ with $BaTiO_3$ significantly improves CTE matching. To validate this, we perform a switching-based power-cycling test, the widely used method for qualifying power module reliability[77]. As detailed in Supplementary Section S16, the sub-module C exhibits minimal parametric shift after 10,000 power-cycling cycles, whereas sub-module A exhibits pronounced forward voltage degradation after ~1200 cycles.

## Discussion

This work demonstrates a multi-die, device-package co-optimized $Ga_2O_3$ module capable of continuous pulsed power switching at 1000 A, 1000 V with fast switching speed and minimal reverse recovery, setting a record $P_C$ among UWBG power devices. A high-$\kappa$ interface design is deployed to replace the conventional post in the JSC package, resulting in over 50% reduction in thermal resistance while enabling >2 kV avalanche breakdown operation at temperatures up to 250 °C. Leveraging the high $C_v$ of $Ga_2O_3$, the intrinsic thermal stability of $Ga_2O_3$ devices, and the electrothermally co-optimized package design, the $Ga_2O_3$ module surpasses Si and WBG counterparts in power capacity density. These findings offer guidance for material selection and packaging strategies in high-power pulsed applications and provide a valuable reference point for module development and power scaling across emerging WBG and UWBG materials.

Looking forward, further upscaling of $Ga_2O_3$ devices and modules toward higher power capacity will require concurrent advances in current and voltage ratings, as well as high-voltage reliability at temperatures beyond the 250 °C demonstrated here. From a material perspective, this calls for the development of larger-diameter $Ga_2O_3$ wafers with improved uniformity and reduced defect density, together with thicker and low-doped epitaxial layers to support higher voltages. At the device and module levels, continued optimization of edge terminations and heterogeneous interfaces in both devices and packaging will be essential to alleviate degradation mechanisms that are accelerated by the combined effects of high electric field and elevated temperature. Beyond diode demonstrations, the realization of $Ga_2O_3$ transistors that leverage the device-package co-design principles identified in this work will be critical for enabling fully integrated, all-$Ga_2O_3$ pulsed-power systems.

## Methods

### Epitaxial structure and device fabrication

The $Ga_2O_3$ wafer consists of a 10-µm Si-doped drift layer with an electron concentration of $1.5 \times 10^{16}$ cm$^{-3}$ grown by hydride vapor phase epitaxy (HVPE) on a conductive Sn-doped (001) β-$Ga_2O_3$ substrate. The schematic of device fabrication steps is shown in Supplementary Section S4. The device fabrication started with the substrate cleaning via ultrasonic treatment in acetone and alcohol soaking. Then, the $Ga_2O_3$ epi-wafers were annealed at 500 °C under an $O_2$ ambient. A Ti/Au (20/80 nm) Ohmic contact was formed as the cathode, followed by rapid thermal annealing. Subsequently, by using a bi-layer photoresist, multi-layer NiO JTE was deposited on the $Ga_2O_3$ drift layer by RF magnetron sputtering technique at room temperature. The NiO layers have varied lengths to enable a graded decrease in effective charge density away from the main junction. The target used in magnetron sputtering was high-purity (99.99%) NiO ceramics. The thickness of each graded p-NiO JTE layer and top p$^+$-NiO layer is 100 nm. The acceptor concentration of p-NiO and p$^+$-NiO is $1.7 \times 10^{17}$ cm$^{-3}$ and $2 \times 10^{19}$ cm$^{-3}$, respectively, as modulated by the $O_2$/Ar gas ratio in the sputtering process. To achieve good anode Ohmic contact on p-NiO, in

situ sputtering of a metal Ni layer (100 nm) is performed in Ar ambient at room temperature utilizing high-purity Ni metal target. The device active area is $3 \times 3$ mm$^2$.

### Device packaging

Schematic of the packaging steps and optical images of three sub-modules are shown in Supplementary Section S5. For sub-modules A and C, 300 nm $SiO_2$ and 300 nm $BaTiO_3$ were deposited on the $Ga_2O_3$ die surface by plasma-enhanced chemical vapor deposition and magnetron sputtering, respectively. The contact window was opened by dry etch for $SiO_2$ and by a lift-off process for $BaTiO_3$. Then top anode and bottom cathode were further finished with an Au/Ti (200 nm) bilayer. The Ti barrier layer can avoid the diffusion of Ag atoms into electrode during the high-temperature silver sintering process. Then the device die attach on the DBC substrate and post attach on the die were formed by silver sintering. A laser-cut mask was used to stencil-print a 50-µm thick silver-sintering paste (provided by NBE Technologies) on die anode. The sintering temperature profile was from room temperature to 230 °C at a ramp rate of 6 °C/min and held at 230 °C for 1 h, followed by air cool to room temperature.

For sub-module B with a post interface, a Mo-Cu post was sintered on the anode; then anode post was sintered on DBC. The subsequent steps, e.g., cathode and stacker-side sintering, soldering and ribbon formation, are the same for all sub-modules. In the soldering process, a solder paste (Sn-42Pb-8Bi) was used for the silver ribbon attach. Before packaging, AlN DBC substrate (provided from Rogers Corporation) was patterned. Firstly, the DBC substrate was covered by Kapton tape as a hard mask, then laser cutting was used to remove the tape outside the pattern, followed by spray etching with $FeCl_3$ solution. The sub-module was encapsulated before thermal resistance measurement and circuit test. In the encapsulation process, a 3-D printed housing was glued to the DBC using the slow-cure epoxy resin (from Bob Smith Industries), then silicone gel (Wacker 612 A/B) was mixed and vacuumed to remove all air bubbles, which was poured into the housing to encapsulate all the components. Finally, the housing was placed on a hotplate and cured at a temperature of 150 °C for 30 min.

### SEM characterization

The SEM image shown in Fig. 1h was captured by a ThermoScientific Helios 5 UC dual-beam workstation that combines a high-resolution scanning electron microscope (SEM) and a focused ion beam (FIB). The measured SEM images were processed with Photoshop software to add false color to highlight different layers of device structure.

### Device static temperature-dependent electrical characterizations

The temperature-dependent forward and reverse I-V measurements were performed by a Keysight B1505A Power Device Analyzer. The probe station model is FormFactor Summit 11000, equipped with a thermal chuck and temperature controller, which supports high-voltage and high-current testing at temperatures from −40 °C to 300 °C. I-V sweeps were conducted using the N1265A high current and high-voltage SMU four-wire Kelvin connection is utilized in the I-V sweep to eliminate the impact from probe-associated parasitic resistance.

### TCAD device simulations

The Technology Computer-Aided Design (TCAD) device simulations were performed using Silvaco to obtain the electric field distribution. The major material parameters and physical models used in the simulation are described in Supplementary Section S6. Appropriate physical models, such as Chynoweth's impact ionization model, field-dependent mobility, Shockley-Read-Hall recombination, Auger recombination, and bandgap narrowing effects, were deployed. Poisson's equation and carrier transport equations were solved self-consistently under steady-state and transient bias conditions. Mesh

refinement was applied in high-field regions and various interfaces to ensure numerical convergence.

## Thermal impedance characterizations

The junction-to-case thermal impedance ($Z$) was measured using the transient dual interface method (TDIM), following the JEDEC 51-14 standard. In this technique, the device under test was subjected to a controlled power step, and the resulting transient temperature response was monitored. Measurements were performed using an Analysis Tech Phase 12 Semiconductor Thermal Analyzer, which provides high temporal resolution for capturing the heating and cooling transients. The thermal impedance curve was derived by deconvolution of the temperature response and applied heating power, yielding the cumulative structure function. $Z$ was determined as the thermal resistance increment between the two test configurations: one with a high-conductivity thermal interface material (TIM) applied between the device case and the cold plate, and the other with a low-conductivity TIM. The key equation governing the transient response is: $Z = (T_j(t) - T_{ref}(t))/P$, where $T_j(t)$ is the measured instant junction temperature, $T_{ref}(t)$ is the reference temperature at the cold plate, and $P$ is the applied heating power. The intersection point of the differential structure functions from both configurations defines the effective junction-to-case thermal resistance.

## Circuit-level characterizations

The short-pulse power switching demonstration, surge current tests with variable pulse widths, and the dynamic switching test for simulation calibration were carried out by a customized circuit platform. Schematics of test circuits, working principle, waveforms, and thermal images of the dynamic test circuit are presented in Supplementary Section S9. Schematics of test circuit, working principle, and waveforms for short-pulse power switching are presented in Supplementary Section S11. Schematics of test circuit, working principle, and waveforms for surge current tests are presented in Supplementary Section S13.

## ANSYS simulation and thermal calibration with experiments

ANSYS Workbench (2021 R1) software was used for package and module simulations in this work. The detailed material parameters, physical model, and calibration with experiments for the steady-state and transient thermal simulation in Figs. 3–5 are described in Supplementary Section S9. The 3D physical structure of the $Ga_2O_3$ packaged device was first constructed in SolidWorks and then imported into ANSYS Workbench, where SpaceClaim was used to simplify the module geometry and define Icepak-recognizable objects such as solids, heat sources, and boundaries. After preprocessing, the model was transferred into Icepak. Thermal properties, material properties, power dissipation, and convection boundary conditions were specified. To ensure accuracy, the computational mesh was refined in regions with steep thermal gradients, such as junctions and interfaces. Steady-state or transient thermal simulations were carried out, and the resulting temperature distribution inside the module structures provided insight into the module's thermal behavior and heat dissipation capability.

## Data availability

All data that support the findings of this study are available in the manuscript and its Supplementary Information file. All data generated in this study have been deposited in the Figshare database. Source data are provided with this paper.

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

## Acknowledgements

We thank NBE Technologies for providing technical support and materials related to the nanosilver paste, and Rogers Corporation for their support in supplying the AlN DBC substrates. We also thank the collaboration with Silvaco for device simulation and the collaboration with Novel Crystal Technologies on Ga$_2$O$_3$ wafer design and epitaxy. J.Y. acknowledges the funding support of the National Natural Science Foundation of China (Grant No. 62425403 and Grant No. 62234007). Y.Z. acknowledges the funding support from the HE Fellowship program and the Centre for Advanced Semiconductors and Integrated Circuits (CASIC) Industry Consortium.

## Author contributions

Y.Z., J.Y., and G.L. conceived the study and supervised the project. H.G. and X.Y. contributed equally to this work. H.G., J.Y., R.Z., and H.W. carried out the device fabrication, characterization, and data analysis. X.Y., Z.Y., M.P., H.C., Y.Q., and D.D. conducted the electrical performance testing and data analysis. H.G., B.W., Z.Z., Q.Y., and G.L. performed the module packaging and data analysis. H.G., X.Y., and Y.Z. wrote and revised the manuscript, and all authors discussed the results and commented on the manuscript.

## Competing interests

The authors declare no competing interests.
