## [Transparent Peer Review file · Nature Communications]

A megawatt ultra-wide bandgap semiconductor module for pulsed power electronics

Corresponding Author: Professor Yuhao Zhang

Version 0:

Reviewer comments:

Reviewer #1

(Remarks to the Author)

The paper addresses the interesting topic of future pulsed power electronics. It mainly concerns the use of Ga₂O₃, an UWBG material addressed in numerous recent publications.

The presented diode Ga₂O₃ module has very good performance.

The originality of this paper comes from the pulsed power application and thus the major drawback of Ga₂O₃, its low thermal conduction, is less significant

However, some points must be better explained for the interest of most readers.

Page 4, concerning Fig. 1b “Pulsed power applications, material selection, as well as device and package designs.”, it is written, “Currently, pulsed power switches fall into two main categories: gas discharge switches and solid-state switches”. However, Fig 1b shows controlled devices like thyatrons or SiC MOSFETs while the paper shows the performance of a Power Diode Module.

So at least some explanations about the role of diodes and controlled devices is expected.

Particularly, what is the application of a good power diode module based on Ga₂O₃, as presented in the paper if no equivalent controlled switched exists?

Furthermore, Fig. 4a shows the schematic of the circuit used to test the surge power capability of the proposed Ga₂O₃ diode module. This circuit uses some SiC MOSFETs. So a reader could understand that SiC MOSFETs are convenient for pulsed power applications, even to avoid the use of power diode modules as in a synchronous converter.

So why a power diode module is very important for pulsed power applications ?

Page 5, it is written :

“As thermal diffusion is negligible, an approximate model dictates the local temperature rise ΔT to scale with $Q / (Cv V)$, where Q is the heat generation energy, and V is the material volume. Hence, materials with higher Cv can absorb more energy with smaller temperature increases, offering superior thermal buffering against fast surges”

This is true, and fig 2.c shows a better, about a ratio of 2, Cv for Ga₂O₃ than for SiC, but the build-in potential is higher in Ga₂O₃ (~ 4.5 V) than in SiC (~2.8 V). So heat generation could be higher for Ga₂O₃. Furthermore, in pulse power applications, heat generation is related to the high-pulsed current.

This issue is no more included in the Baliga's FOM. So the conclusion of such data is not so easy and demands further discussion.

Fig. 2a, Breakdown voltage enhancement and sub-modules' electrical characteristics.

The figure seems to be related to a heterojunction but it is not mentioned in the caption. It will be clearer for most readers to explicitly mention it in the caption.

Concerning the presented result, it is written, “all simulated under a reverse bias of 2000 V”, while in Fig 2b), “Reverse I-V characteristics of three types of Ga₂O₃ sub-modules with different package structures” show VBR about of 930 V, 1560 V for structures A and B, so this is inconsistent. That needs some explanations or corrections.

Moreover Fig. 2a shows a higher electric field (with orange region) in case of sub-modules A and C but not in case of sub-module B. This is also inconsistent with the I-V characteristic in Fig 2b.

So please correct this or give an adequate explanation.

Furthermore, on page 6 it is written, "the VB increases from 2010 V to 2240V with T from RT to 250 °C, exhibiting a positive temperature coefficient of ~1 V/°C, which suggests the avalanche capability."

It is true, but Fig. 2d also shows a degradation of VBR (VB) from 2240 V to about 1300 V from 250 °C to 300 °C. Is there a clear explanation ? Perhaps this strange behavior comes from some degradation of insulating layers... That demands some explanations and that suggests the difficulty of reaching 250 °C.

In page 6, comes some comments on Fig. 3. In the caption it is written,

"the insets show the optical images of the two prototyped submodules, where encapsulation is omitted to more clearly reveal the package structure.". That's a good idea.

Moreover in Supplementary Section S5 – Package Process, in the encapsulation section, it is written, "A 3D-printed polymer housing was glued to the DBC surface using a slow-curing epoxy resin (Bob Smith Industries). A two-part silicone gel (Wacker 612 A/B) was mixed, vacuum-degassed to eliminate air bubbles".

Unfortunately, there is no presentation of the housing. The polymer structure, as well, silicone gel are some limitations for an industrial product at 250 °C peak temperature. At least some comments or explanations must be given. For instance such housing is needed to protect the module against moisture and other environmental duties.

In Supplementary Section S7 - Temperature-dependent Breakdown Characterization of SiC SBD, it is shown the temperature limitation of the commercial SiC SBD (ROHM, S6304).

The comparison is not fair because some research papers show better performance even for high-voltage devices [1].

In Supplementary Section S9 - ANSYS Workbench simulation and calibration with experimental, Figure S9-2a is confusing because a similar but different circuit is presented in Fig. 4a and the associated optical pictures are the same.

Some explanation must be given, at least for Fig. 4a, in the main text, where the SiC MOSFETs role must be given and the references of the devices are needed. Some explanations exist in Supplementary Section S11 –Experimental circuit testing setup for short-pulse DPT but the MOSFETs references are not given. Moreover, this also contributes to the idea that SiC MOSFETs can replace Ga2O3 diodes in a pulsed power application. Further explanation is needed.

In figure S14, the color scale is not readable. Please correct that.

Fig. 3 | Thermal impedance characterization and time-resolved maximum power capacity derivation.

In Fig. 3e and Fig. 3f, it is clearly shown that the Ga2O3 module yields a better Zth or maximum transient power capacity density, respectively. However, this is not entirely fair because the SiC diode uses a TO220 package and has a low current rating, 19 A. It is even possible to deduce that, to better respect the maximum power density need in a pulse power application, it is sufficient to choose a more recent diode with a higher current rating. So at least some additional explanations must be given.

Fig. 4 | Demonstration of megawatt pulsed power switching in practical power converters.

Concerning this figure, d) simulated time-resolved Tj evolution of the Ga2O3 sub-module with high-κ interface, but the associate figure shows two curves. The read one (257 A) has a written label "breakdown" without any explanation. Some explanations or corrections must be given.

Fig. 5 | Surge current capability of Ga2O3 sub-module in conduction-dominant pulsed power applications.

In the caption, it is written, "... tested with an increased surge energy at a fixed pulse width of 3.6 μs. The critical peak surge current is over 800 A." For the top curve, it is written "device degradation", but no explanation is given. Please correct this.

Concerning the "Methods section", it's rather confusing because the purpose of this section, after the conclusion, is unclear. Probably, a section before the conclusion with a small introduction to explain the purpose of the section will be clearer.

The conclusion is very positive for the performance of Ga2O3 modules for "pulsed power switching at 1000 A, 1000 V".

However, an analysis of the road map to industrial products should be very interesting for most readers. For instance in [2] it is written (by some of the same authors), "We envision the following immediate research gaps that need to be addressed for advancing Ga2O3 into industrial power electronics applications".

Probably, a similar analysis of pulse power applications will be very useful.

Conclusion: the paper needs major revisions.

References

[1] KIMOTO, Tsunenobu, YAMADA, Kyosuke, NIWA, Hiroki, et al. Promise and challenges of high-voltage SiC bipolar power devices. *Energies*, 2016, vol. 9, no 11, p. 908.

[2] QIN, Yuan, WANG, Zhengpeng, SASAKI, Kohei, et al. Recent progress of Ga₂O₃ power technology: large-area devices, packaging and applications. *Japanese Journal of Applied Physics*, 2023, vol. 62, no SF, p. SF0801.

Reviewer #2

(Remarks to the Author)

The proposed methodology is novel and well-executed, integrating device-level dielectric engineering with packaging-level electrothermal co-optimization. The incorporation of a high-k BaTiO₃ interface for field flattening, combined with junction-side cooling and optimized die attach design, effectively address high electric field and thermal management challenges. The authors also demonstrate a deep understanding of Ga₂O₃'s intrinsic properties, particularly its high volumetric heat capacity and superior high temperature stability, and skillfully design both the device and packaging to exploit these advantages for the high-pulse power applications. The module level results are encouraging and support the technical soundness of the concept. Overall, this thoughtful UWBG co-design represents a promising step toward the practical deployment of Ga₂O₃ based modules, particularly for high pulse energy use.

Points for clarification / potential concerns:

1. It would be beneficial to add some comments on the dipole properties of BTO and why this material is the superior choice to others that may also possess such properties. I.e., why BTO?

2. Will the high-k BaTiO₃ interface layer functionally replace the conventional SiO₂/SiN₄ surface passivation on chips, or is it intended to be added on top of the conventional passivation to act as an additional field-modulation layer? If it is a replacement, please comment on its insulating reliability and long-term stability? How will it stand up to humidity, a known issue with BaTiO₃? I.e., will the field dampening properties be sacrificed? If so, by how much?

3. Given the CTE mismatch between BaTiO₃ and Ga₂O₃, how robust is the interface during repeated thermal cycling and high-field pulsed operation? Please comment on potential interfacial defects, charge trapping/polarization drift, breakdown-voltage stability, and any signs of delamination or micro-cracking.

4. BaTiO₃ is deposited by RF magnetron sputtering in an Ar/O₂ ambient. How compatible is this process with large-area fabrication and downstream metallization/packageing?

5. Can the high-k interface be extended to other WBG/UWBG devices (e.g., SiC, GaN, diamond)? If so, please outline material-specific adaptations, interface reliability under cycling, CTE matching, and process compatibility.

Reviewer #3

(Remarks to the Author)

Reviewer #4

(Remarks to the Author)

This is an excellent manuscript highlighting the performance of Ga₂O₃ UWBG power semiconductor devices, especially for pulsed power applications. The authors provided a comprehensive literature review, indicating the unique challenges and how Ga₂O₃ can offer potential solutions. The emphasis on the electro-thermal co-design approach is a new trend for all future power electronic systems. The experimental demonstration of the Mega Watt pulse power is also impressive, completed with device simulation, packaging design, electrical and thermal considerations. I truly enjoy reading this paper.

Perhaps one comment that may help further improve this paper is to discuss how the repetition rate (or duty cycle) have on the thermal capacity of the proposed power module. Will the proposed pulse power application require any thermal exchanger to further improve the performance?

Reviewer #5

(Remarks to the Author)

The paper demonstrates the first ultra-wide bandgap (UWBG) power module, which enhances the power capacity of UWBG devices by over three orders of magnitude and identifies pulsed power electronics as a novel application for Ga₂O₃ power devices. Considering the rapidly growing interest in UWBG power electronics and huge industrial investment in this domain, such a performance breakthrough is expected to have significant impact. This work also presents fundamental understanding by providing a high-k device-package interface and highlighting the significance of heat capacity as a critical material attribute for pulsed power electronics, suitable to various different semiconductors. This work could attract widespread attention from this perspective.

The manuscript is clear, well written and well structured. The data are comprehensive, covering device, packaging, and

circuit-level verification, supported by theoretical analysis. To further improve the paper, the following technical points should be clarified:

1. For experimental circuit tests, is it possible to measure the power module under higher slew rate and switching frequency? The authors are also suggested to comment on the limiting factors for slew rate and frequency in this circuit test.
2. Ga₂O₃ devices are known to be promising for high-voltage applications, with devices up to 10,000 V having been experimentally reported. Does the superior power capacity of Ga₂O₃ retain for higher-voltage devices? The authors may add supplemental simulations.
3. Following the above point, please also analyze if the power capacity advantage over SiC counterparts retains at higher voltage, as SiC modules have been demonstrated up to 10 kV.
4. In Fig. 1f, please clarify the physical mechanism by which the high- κ dielectric introduces dipole polarization charges at both the high- κ /metal and high- κ /Ga₂O₃ interfaces.
5. In Fig. 3b, the peak temperature appears at the anode edge for submodule B, but at the device center for submodule C. Please explain the difference.
6. Oscillations are observed in both the voltage and current waveforms in Fig. 4f and Fig. 4g following the switching event. Please explain the origin of these oscillations. Additionally, are there any methods that can be applied to mitigate this phenomenon?
7. For the pulse power application discussed, what is the rationale for selecting a pulse width of 3.6 μ s? Furthermore, is it feasible to reduce the pulse width?

##

Reviewer #6

(Remarks to the Author)

+ What are the noteworthy results?

- This manuscript presents a novel power module based on Ga₂O₃ power devices for pulsed power applications. The module under investigation promises enhanced current/voltage capability as well as careful temperature management. The Authors also assembled a multi-die module allowing switching at 1kA/1kV.

+ Will the work be of significance to the field and related fields? How does it compare to the established literature? If the work is not original, please provide relevant references

- This work is significant to the field, as the proposed sample shows unprecedented features.

+ Does the work support the conclusions and claims, or is additional evidence needed?

- No additional evidence is needed.

+ Are there any flaws in the data analysis, interpretation and conclusions? Do these prohibit publication or require revision?

- Some minor remarks follow:

In fig. 1b there is a typo: "shot" is written but "short" was intended;

Initially, the Authors claim that the value of the thermal conductivity of the assembly materials is not relevant in the case of pulsed power electronics applications, whereas the heat capacity is the most important parameter; however, in the "Device-package electrothermal co-optimization" paragraph, they underline the low- κ t value of Ga₂O₃. Fig. 5h also shows that κ t plays a relevant role in determining the thermal behavior of the modules. More details are needed to evaluate the individual impact of the two above parameters on the FOMs of the proposed module.

+ Is the methodology sound? Does the work meet the expected standards in your field?

- The methodologies are sound and meet the standard.

+ Is there enough detail provided in the methods for the work to be reproduced?

- The article as well as the supplementary material provide enough detail for the work to be reproduced.

Version 1:

Reviewer comments:

Reviewer #1

(Remarks to the Author)

The paper addresses the interesting topic of future pulsed power electronics. It mainly concerns the use of Ga₂O₃, an UWBG material addressed in numerous recent publications.

The presented diode Ga₂O₃ module has very good performance.

The originality of this paper comes from the pulsed power application and thus the major drawback of Ga₂O₃, its low thermal conduction, is less significant.

All of my comments have been correctly even acceptably addressed.

The methodology and presentation of the results is sufficiently rigorous.

Conclusion: the paper is ready for publication.

Reviewer #2

(Remarks to the Author)

None at this time.

Reviewer #3

(Remarks to the Author)

Reviewer #5

(Remarks to the Author)

The authors have properly addressed all of my previous comments. The revised manuscript now provides a clear and well-supported explanation of electric field management improved by high permittivity dielectrics, along with an enhanced analysis of thermal behavior under short pulse conditions. The modifications address my concerns effectively.

I have no additional comments and accept the publication.

Reviewer #6

(Remarks to the Author)

The Authors have addressed my comments.

Response Letter (Manuscript ID: NCOMMS-25-66126)

We would like to thank Reviewers for their constructive and helpful suggestions. In the *Response Letter*, all comments from reviewers have been addressed point-by-point, and the corresponding revisions have been highlighted in **BLUE** in the revised manuscript. The replies are listed below.

	1. Response to 1 st reviewer's comments	2. Response to 2 nd reviewer's comment
Pages numbers	1~23	24~30
	3. Response to 3 rd reviewer's comments	4. Response to 4 th reviewer's comments
Pages numbers	31	32-34
	5. Response to 5 th reviewer's comments	6. Response to 6 th reviewer's comments
Pages numbers	35-41	42-43

1. Reply to the 1st reviewer's comments

Reviewer #1 (Remarks to the Author):

The paper addresses the interesting topic of future pulsed power electronics. It mainly concerns the use of Ga₂O₃, an UWBG material addressed in numerous recent publications.

The presented diode Ga₂O₃ module has very good performance.

The originality of this paper comes from the pulsed power application and thus the major drawback of Ga₂O₃, its low thermal conduction, is less significant.

However, some points must be better explained for the interest of most readers.

Response: We sincerely thank the reviewer for the positive and constructive comments. We appreciate the acknowledgment of the significance of pulsed power applications and the performance of our Ga₂O₃ diode module. Following the reviewer's suggestions, we have revised the manuscript accordingly and improved the explanations.

Comment (1): Page 4, concerning Fig. 1b "Pulsed power applications, material selection, as well as device and package designs.", it is written, "Currently, pulsed power switches fall into two main categories: gas discharge switches and solid-state switches". However, Fig 1b shows controlled devices like thyatrons or SiC MOSFETs while the paper shows the performance of a Power Diode Module. So at least some explanations about the role of diodes and controlled devices is expected. Particularly, what is the application of a good power diode module based on Ga₂O₃, as presented in the paper if no equivalent controlled switched exists?

Response: We sincerely thank the reviewer for this insightful comment. We fully agree that the manuscript should more clearly describe the roles of both power transistors (e.g., Si IGBT, SiC MOSFETs) and power

diodes in pulsed-power systems.

In pulsed power systems, power transistors are usually used to control the pulse initiation, whereas power diodes are ubiquitously used for controlling the energy flow, as well as providing protection functions during fast charge/discharge events. In nearly all pulsed-power architectures, including Marx generators, linear transformer drivers, capacitor-bank discharges, XRAM current multipliers, and inductive-load pulsers, diodes experience extremely large surge currents during capacitor discharge, very high di/dt from rapid current interruption, high reverse blocking voltage during pulse formation, and often require avalanche and transient energy absorption. These stresses are often more severe than those seen by power transistors (i.e., controlled switches), making the performance of the diode equally critical.

As a general example, many pulsed-power systems drive inductive loads, such as pulsed magnetic-field generators, electromagnetic launchers, MRI gradient coils, pulsed solenoids, and magnetic-forming equipment. In inductive load switching circuit (Fig. R1)¹⁻³, when the main switch turns off, the inductor forces $V=L(di/dt)$, causing dangerous voltage overshoot. Here, the diode must conduct the full inductor current through a freewheeling path, suppress large $L \cdot di/dt$ voltage spikes, undergo controlled avalanche under extreme pulses to protect the switch.

Figure R1. Inductive load switching circuit with freewheeling diode¹⁻³.

For a more specific example, in electromagnetic launching systems (Fig. R2)⁴, the drive capacitor discharges through the switch and coil. Once the capacitor is empty, the switch opens and the coil current must fully flow through the diode until the launch completes. This requires a diode module with kA-level surge capability and strong avalanche robustness.

Figure R2. Single-stage reconnection electromagnetic launch model driving circuit⁴.

As another example, the XRAM current-multiplier circuit (Fig. R3) relies heavily on high-power diodes⁵. In XRAM systems, multiple inductors are first charged in parallel by a low-voltage source and then discharged in series to generate an exceptionally large load current. During this series-discharge process, the

diodes must steer and combine the inductor currents, block high reverse voltages, and withstand extreme surge stress. As a result, XRAM architectures impose exceptionally demanding requirements on the diodes, with peak currents commonly reaching the kilo-ampere range and often exceeding 3 kA⁵.

Figure R3. XRAM current multiplier circuit⁵.

As an example for the isolation function, some high-voltage pulse circuits often impose very high transient reverse blocking demand on diodes. In voltage-multiplying and pulse-forming systems, such as Marx generators and stage-charging networks, diodes must block multi-kilovolt reverse voltage, high-frequency charging transients, and echo/reflection voltages from pulse lines. For instance, Marx generator charging stages shown in Fig. R4 require series-connected diodes to provide isolation and prevent reverse conduction during triggering⁶.

Figure R4. Marx generator circuit⁶.

Moreover, we would like to emphasize that many industrial pulsed-power circuits do not necessarily rely on controlled switches. In numerous practical systems, power diodes alone form the core functional element, and no transistor is needed. For example, space and communication systems widely employ Capacitor-Diode-Based Voltage Multipliers (CDVMs), and microelectronic applications such as RF passive transponders, passive wireless microsensors, and low-power portable devices commonly use Cockcroft-Walton (CW) multiplier stages (Fig. R5) for voltage boosting^{7,8}. These circuits operate purely through the coordinated action of diodes and capacitors, without any gate-driven switches, enabling exceptionally simple, lightweight, and high-reliability implementations.

Similarly, high-voltage generators used in X-ray sources, electrostatic precipitators, HV capacitor chargers, plasma generators, and many other pulsed-power systems rely heavily on CW-type multipliers. In

such systems, the diode must sustain high reverse voltage, fast transient charging current, and low reverse-recovery loss to maintain voltage regulation and pulse fidelity. These industrial applications therefore place stringent requirements on diode performance, independent of any controlled switching device.

Figure R5. Cockcroft-Walton voltage multiplier^{7,8}.

Overall, a high-voltage, high-current, high-temperature-operational, and avalanche-and-surge capable Ga₂O₃ diode module can serve as a fundamental building block in a wide range of pulsed-power and high-voltage systems. This further underscores the practical value and immediate applicability of the Ga₂O₃ diode module demonstrated in our work.

Corresponding change in manuscript: Yes

Location of Change: The revisions have been incorporated into the 2nd paragraph of the “Pulsed power applications and material selection” section (Page 4) in the revised manuscript.

Revised text:

“Currently, pulsed power devices fall into two main categories: gas discharge devices and solid-state devices (Fig. 1b)³⁰.”

Expanded paragraph added:

“Next-generation pulsed power devices, including both controlled switches (i.e., power transistors) and power diodes, are therefore in high demand. In pulsed-power systems, transistors are often used for initiating and shaping the pulse, whereas power diodes play indispensable roles in energy steering, stage isolation, rapid charge management, and transient overvoltage suppression. For instance, in inductive-load pulsed circuits, the diode must withstand and conduct large surge currents while clamping voltage spikes induced by rapid current transients ($L \cdot di/dt$), all within several micro-seconds. In high-voltage pulse generation circuits, such as Marx and Cockcroft-Walton generators, the diode is further required to reliably block kilovolt reverse voltages.”

References:

[1] Williams, B.W. (1987). Load Considerations. In: Power Electronics. Palgrave, London. https://doi.org/10.1007/978-1-349-18525-2_6.
 [2]<https://www.analog.com/en/resources/technical-articles/switching-inductive-loads-with-safe-demagnetization.html>.
 [3]<https://www.powerelectronicstips.com/what-is-the-purpose-of-a-freewheeling-diode-in-power-electronics>.

[4] Dong, L., Jiang, W., Xie, X. et al. Design of maglev train driven by reconnection electromagnetic launcher. *Sci Rep* 15, 22957 (2025). <https://doi.org/10.1038/s41598-025-05992-0>

[5] H. Li et al., "A Repetitive High-Current Pulse Generator Circuit Based on Multistage Pulse Transformers," in *IEEE Journal of Emerging and Selected Topics in Power Electronics*, vol. 9, no. 3, pp. 3189-3200, June 2021, doi: 10.1109/JESTPE.2020.3010132.

[6] Y. Wu, K. Liu, J. Qiu, X. Liu and H. Xiao, "Repetitive and High Voltage Marx Generator Using Solid-state Devices," in *IEEE Transactions on Dielectrics and Electrical Insulation*, vol. 14, no. 4, pp. 937-940, Aug. 2007, doi: 10.1109/TDEI.2007.4286529.

[7] Khan, K. L., Verma, R., & Roy, A. (2024). Review of High Voltage Pulsed Power Supplies and Power Electronics in Pulse Power Generation. *Power Research - A Journal of CPRI*, 19(2), 157–162. <https://doi.org/10.33686/pwj.v19i2.1140>

[8] F. Yuan and N. Soltani, "Design techniques for power harvesting of passive wireless microsensors," 2008 51st Midwest Symposium on Circuits and Systems, Knoxville, TN, USA, 2008, pp. 289-293, doi: 10.1109/MWSCAS.2008.4616793.

Comment (2): Furthermore, Fig. 4a shows the schematic of the circuit used to test the surge power capability of the proposed Ga₂O₃ diode module. This circuit uses some SiC MOSFETs. So a reader could understand that SiC MOSFETs are convenient for pulsed power applications, even to avoid the use of power diode modules as in a synchronous converter. So why a power diode module is very important for pulsed power applications?

Response: We thank the reviewer for raising this important point. Building upon the clarification provided in our **Response to Comment (1)**, power transistors and diodes are both required in pulsed power system. The controlled switches trigger the pulse, while power diodes perform essential functions such as energy steering, stage isolation, rapid charge management, and transient overvoltage suppression. In this test circuit of Fig. 4a, the SiC MOSFETs' only purpose is to control short-pulse, high-voltage, and high-current waveforms. These pulses are then delivered to the Ga₂O₃ diode module, which is the only rectifying element in the experiment – their functionalities and implementations are independent (e.g., can from two different semiconductor technologies).

Although SiC MOSFETs include a body diode and may be used in synchronous rectification, neither the MOSFET body diode nor synchronous operation can replace a dedicated high-power diode in pulsed-power architectures, for the following reasons: 1) In any switching stage, a finite dead time is required to avoid cross-conduction. During this interval: the energy must flow through a diode path, and the MOSFET's body diode becomes the unintended conduction path. Using only the MOSFET body diode leads to large reverse-recovery charge, high reverse current overshoot, elevated switching losses, and severe voltage overshoot during $L \cdot di/dt$ events⁹. Thus, a separate external diode with fast recovery and strong robustness is essential, especially under high surge-current conditions used in our tests. 2) The SiC body diode typically suffers from high forward voltage, large reverse-recovery charge, and strong bipolar degradation under high-

current stress, poor avalanche and surge ruggedness compared with dedicated power diodes¹⁰. These limitations become critical in pulsed-power systems. Thus, a discrete diode (module) is required to protect the MOSFET and maintain pulse fidelity.

Finally, regarding the material choice, we agree that SiC has excellent commercial diodes, as shown in Fig. 3f in our manuscript, Ga₂O₃ diodes offer unique and complementary advantages, higher blocking strength, higher transient power capability, and superior high-temperature performance compared with SiC diodes, making them highly suitable for next-generation pulsed-power applications. Although commercial Ga₂O₃ transistors are still in their early stages, substantial progress is occurring toward Ga₂O₃ power transistors that would eventually complement the diode demonstrated in this work. Recent reports present the Ga₂O₃ transistors with super high voltage of 10kV, which presents great potential in the power applications. Once Ga₂O₃ transistor technology is mature, a Ga₂O₃ diode module will be the natural complementary component, enabling fully Ga₂O₃-based pulsed-power platforms with unprecedented power density and compactness.

Corresponding change in manuscript: No

Reference:

- [9] X. Liu, X. Li, C. Herrmann and T. Basler, "The Impact of the Dead-Time on the Reverse Recovery Behavior of SiC-MOSFET Body Diodes," 2023 35th International Symposium on Power Semiconductor Devices and ICs (ISPSD), Hong Kong, 2023, pp. 322-325, doi: 10.1109/ISPSD57135.2023.10147719.
- [10] X. Jiang et al., "Investigation on Degradation of SiC MOSFET Under Surge Current Stress of Body Diode," in IEEE Journal of Emerging and Selected Topics in Power Electronics, vol. 8, no. 1, pp. 77-89, March 2020, doi: 10.1109/JESTPE.2019.2952214.

Comment (3): Page 5, it is written: "As thermal diffusion is negligible, an approximate model dictates the local temperature rise ΔT to scale with $Q/(C_v V)$, where Q is the heat generation energy, and V is the material volume. Hence, materials with higher C_v can absorb more energy with smaller temperature increases, offering superior thermal buffering against fast surges"

This is true, and fig 2.c shows a better, about a ratio of 2, C_v for Ga₂O₃ than for SiC, but the built-in potential is higher in Ga₂O₃ (~4.5 V) than in SiC (~2.8 V). So heat generation could be higher for Ga₂O₃. Furthermore, in pulse power applications, heat generation is related to the high-pulsed current.

This issue is no more included in the Baliga's FOM. So the conclusion of such data is not so easy and demands further discussion.

Response: We appreciate the reviewer's insightful comments regarding the relationship between heat generation, built-in potential, and the thermal response during fast surge events. Our original intention was to provide a generic discussion on the transient temperature rise under the assumption of comparable heat generation (i.e., power loss). Our comparison of transient thermal buffering focuses on the material-dependent volumetric heat capacity C_v , assuming a similar heat generation level. Meanwhile, we agree that

the built-in potential and current amplitude affect the actual power dissipation, and we have clarified this section accordingly.

Although the intrinsic built-in potential of Ga₂O₃ (~4.5 V) is higher than that of SiC (~2.8 V), the effective forward voltage during conduction is also determined by device structure interface properties. For example, in our NiO/Ga₂O₃ heterojunction diode, the measured V_{ON} is ~2 V, significantly lower than the theoretical built-in potential of Ga₂O₃. In contrast, commercial SiC devices (e.g., the body diode of SiC MOSFETs) often exhibit a similar or even higher V_{ON} (which is usually close to SiC's built-in potential). Thus, the practical forward conduction loss of Ga₂O₃ diodes is not necessarily larger than that of SiC counterparts. Under high current surge conditions, the forward voltage is dominated by resistive drop rather than the built-in voltage. Due to the diverse selection of device structure (e.g., homo-junction or hetero-junction) and the possibility of varying operation conditions (e.g., high surge current), it is very difficult to establish a point-to-point relation between a material's built-in potential and the conduction loss of power diodes.

We appreciate the reviewer's suggestion and have revised the wording to avoid implying that heat generation (i.e., power loss) is assumed to be identical across materials for the discussion under the specific context. This revision clarifies that the comparison is made under identical heat-load conditions and highlights the physical meaning of C_v without conflating it with power loss mechanisms.

Corresponding change in manuscript: Yes

Location of Change: The revisions have been incorporated into the 3rd paragraph of the "Pulsed power applications and material selection" section (Page 5) in the revised manuscript.

Revised text:

"Hence, a higher C_v indicates a smaller temperature increase under the identical power loss³²."

Comment (4): Fig. 2a, Breakdown voltage enhancement and sub-modules' electrical characteristics.

The figure seems to be related to a heterojunction but it is not mentioned in the caption. It will be clearer for most readers to explicitly mention it in the caption. Concerning the presented result, it is written, "all simulated under a reverse bias of 2000 V", while in Fig 2b), "Reverse $I-V$ characteristics of three types of Ga₂O₃ sub-modules with different package structures" show V_{BR} about of 930 V, 1560 V for structures A and B, so this is inconsistent. That needs some explanations or corrections.

Moreover Fig. 2a shows a higher electric field (with orange region) in case of sub-modules A and C but not in case of sub-module B. This is also inconsistent with the $I-V$ characteristic in Fig 2b. So please correct this or give an adequate explanation.

Response: We sincerely thank the reviewer for carefully examining Fig. 2. We provide detailed clarifications below and expand the discussion from the device-physics perspective to ensure full transparency. As suggested, the caption of Fig. 2a has been revised to: "Simulated two-dimensional electric field contours in JSC-packaged NiO/Ga₂O₃ heterojunction devices." This clearly informs readers that the sub-modules correspond to NiO/Ga₂O₃ heterojunction diodes.

We agree that the breakdown voltages (V_B) in Fig. 2b (930 V for sub-module A and 1560 V for sub-module B) are lower than the 2000-V bias used in Fig. 2a (so simulation here presents a somewhat ‘hypothetical’ case for these two sub-modules). Our purpose was to compare how different package structures redistribute electric fields under the same applied voltage, which is a standard and widely accepted method in power device research and presented in numerous publications¹¹⁻¹³. Using the same bias condition allows the differences in field modulation between sub-modules A, B, and C to be directly visualized and fairly compared. A unified bias allows readers to see where the weakness of each structure lies, even if that voltage exceeds the measured breakdown of some structures.

The reviewer observed that peak electric fields in Fig. 2a appear higher in sub-modules A and C, yet their measured breakdown voltages differ in Fig. 2b. We emphasize that: V_B depends not only on peak field magnitude, but also on where the peak field occurs and the critical field of the material at that location. For sub-module A with the lowest V_B of 930V, at 2000 V bias, the peak E -field in submodule A is ~ 19 MV/cm inside SiO_2 , but the color bar saturates at 6 MV/cm to maintain visualization range. The deposited SiO_2 has a critical breakdown field around 10 MV/cm (depending on thickness and quality). Any local concentration >10 MV/cm triggers dielectric failure, forming leakage paths or micro-cracks. The SiO_2 /metal interface is especially sensitive to field crowding, often dominating early failures. Thus, even though the 2000-V simulation is above its measured breakdown voltage, the field distribution correctly reveals that SiO_2 is the limiting factor in sub-module A.

For sub-module B with V_B of 1560V, although submodule B shows a lower peak field (~ 4.4 MV/cm) at 2000 V, the critical point is that the peak field occurs directly in the main $\text{NiO}/\text{Ga}_2\text{O}_3$ junction. Any localized field enhancement in the main junction leads to stronger local heating and faster thermal runaway. Meanwhile, junction is highly sensitive to defect-assisted impact ionization, sputtered NiO contains high density of grain boundaries and vacancies, which will trigger trap-assisted tunneling and localized thermal spots. Thus, sub-module B fails at 1560 V even though its simulated peak field (at 2000 V) is lower than that of sub-module C. For sub-module C with the highest V_B , the peak field of 5.7 MV/cm is located inside Ga_2O_3 at the edge termination, not the main junction with the hetero-interface with NiO . This configuration is superior because high-quality single crystalline Ga_2O_3 has an ultra-high theoretical critical field (>8 MV/cm), safe even at 5.7 MV/cm, and the field in the main junction is only 3.05 MV/cm. The edge termination can maintain uniform E -field and suppress hot spots in the main junction. Thus, sub-module C achieves the highest measured breakdown voltage.

To further verify consistency between the simulated and measured breakdown voltages, we have added electric-field contours simulated at the actual breakdown points. As shown in Fig. R6, at 930 V for sub-module A, the peak electric field within the SiO_2 layer reaches approximately 11.7 MV/cm, which exceeds the dielectric strength of SiO_2 and confirms that breakdown is governed by dielectric failure. For submodule B, at its measured breakdown voltage of 1560 V, the peak electric field at the main junction region is approximately 4.2 MV/cm, which is significantly higher than the average junction field. Given the large device area (3 mm \times 3 mm), localized leakage paths can readily initiate in this high-field region, ultimately

leading to breakdown.

Fig. R6. Simulated two-dimensional electric field contours in JSC-packaged NiO/Ga₂O₃ heterojunction devices (a) with a SiO₂ interface (control sub-module A) at 930 V, (b) with a post interface (control sub-module B) at 1560 V.

Corresponding change in manuscript: Yes

Location of Change:

(1) The caption of Fig. 2a has been revised to: “Simulated two-dimensional electric field contours in JSC-packaged NiO/Ga₂O₃ heterojunction devices.”

(2) Fig. R6. and related explanation have been incorporated into the “**Supplementary Section S6**” section (Page 8) in the revised Supplementary Information.

Reference:

[11] J. Wen et al., "Vertical β -Ga₂O₃ Power Diodes: From Interface Engineering to Edge Termination," in IEEE Transactions on Electron Devices, vol. 71, no. 3, pp. 1606-1617, March 2024, doi: 10.1109/TED.2024.3360016.

[12] Sushovan Dhara, Nidhin Kurian Kalarickal, Ashok Dheenana, Chandan Joishi, Siddharth Rajan. β -Ga₂O₃ Schottky barrier diodes with 4.1 MV/cm field strength by deep plasma etching field-termination. Appl. Phys. Lett. 14 November 2022; 121 (20): 203501.

[13] X. Liu et al., "1.7-kV Vertical GaN-on-GaN Schottky Barrier Diodes With Helium-Implanted Edge Termination," in IEEE Transactions on Electron Devices, vol. 69, no. 4, pp. 1938-1944, April 2022, doi: 10.1109/TED.2022.3153594.

Comment (5): Furthermore, on page 6 it is written, “the V_B increases from 2010 V to 2240V with T from RT to 250 °C, exhibiting a positive temperature coefficient of ~ 1 V/°C, which suggests the avalanche capability.”.

It is true, but Fig. 2d also shows a degradation of V_{BR} (V_B) from 2240 V to about 1300 V from 250 °C to 300 °C. Is there a clear explanation? Perhaps this strange behavior comes from some degradation of insulating layers... That demands some explanations and that suggests the difficulty of reaching 250 °C.

Response: We thank the reviewer for this insightful comment. The observation is correct that while the breakdown voltage (V_B) increases from room temperature to 250 °C with a positive temperature coefficient, Fig. 2d indeed shows a pronounced degradation of V_B when the temperature further increases from 250 °C to 300 °C. This behavior is not contradictory but instead reflects the appearance of a non-ideal preliminary breakdown mechanism at extreme high temperatures, as explained below.

We believe the V_B degradation observed at 300 °C is primarily attributed to the trap-mediated preliminary breakdown at a lower voltage before the intrinsic avalanche breakdown initiates, and such preliminary breakdown is likely due to the trap-assisted tunneling (TAT) current, which is highly dependent on temperature and electric field and has been widely reported as the preliminary breakdown mechanism in wide-bandgap and ultra-wide bandgap power devices¹⁴⁻¹⁸. Moreover, this trap-mediated breakdown exhibits a negative temperature coefficient $\sim 11\text{V}/^\circ\text{C}$ ¹⁹, which is close to the V_B transition from 250 °C to 300 °C in our device.

The location of trap-mediated preliminary breakdown is usually localized and consistent with the peak electric field location^{18,20}. As shown in Fig. R7a, under a reverse bias of ~ 2 kV, the peak electric field is at the Ag/BTO/Ga₂O₃ edge region. As the BTO layer is deposited by magnetron sputtering, which typically contains a non-negligible density of grain boundaries, a relatively high density of interface states can exist at both the Ag/BTO interface and the BTO/Ga₂O₃ interface. According to the band diagram (Fig. R7b), under reverse bias, electrons can tunnel from Ag into the BTO layer via interfacial traps and subsequently tunnel into Ga₂O₃ through defect states at the BTO/Ga₂O₃ interface. This path competes with the usual thermal emission path (see dashed lines in Fig. R7b). At elevated temperatures, this trap-assisted tunneling process is thermally activated and significantly enhanced by high electric field, leading to rapidly increased leakage current and premature breakdown at 300 °C.

In fact, similar trap-assisted preliminary breakdown mechanisms were also observed in other two submodules, whilst the preliminary breakdown locations can be very different. Fig. R8 shows the temperature-dependent I - V characteristics of Ga₂O₃ Sub-module A and Sub-module B, V_B both show negative temperature coefficient throughout temperatures from RT to 300 °C. Specifically, for sub-module A with a SiO₂ interface, the peak electric field is concentrated at the Ag/SiO₂/Ga₂O₃ edge, where tunneling can likewise be assisted by defect states in the SiO₂ interface region. For sub-module B with the post interface, the peak electric field shifts to the main NiO/Ga₂O₃ junction edge, where tunneling is facilitated by defect states in the NiO layer and the heterojunction. In all cases, increasing temperature leads to a sharp rise in leakage current and a substantial reduction in breakdown voltage. Therefore, we believe that the dominant failure mechanism of sub-module C at 300 °C is the trap-assisted preliminary breakdown possibly originated from the interface states between Ag, BTO and Ga₂O₃.

Fig. R7. (a) Simulated two-dimensional electric field contours in JSC-packaged NiO/Ga₂O₃ heterojunction devices with a BTO interface (Sub-module C) at 2 kV. (b) Band diagram of the Ag/BTO/Ga₂O₃ structure under reverse high voltage. The dashed line shows the usual thermal electron emission path, and the solid line shows the TAT leakage path.

Fig. R8. Temperature-dependent reverse I - V characteristics of the Ga₂O₃ sub-module A with SiO₂ interface and sub-module B with post interface.

Corresponding change in manuscript: Yes

Location of Change: The revisions have been incorporated into the 5th paragraph of the “Device-package electrothermal co-optimization” section (Page 7) in the revised manuscript.

Expanded text:

“When the temperature is further increased to 300 °C, V_B decreases to 1330 V, which is likely due to the trap-assisted preliminary breakdown possibly originating from the interface states between Ag, BTO and Ga₂O₃. Such a trap-assisted breakdown has been widely reported in other ultra-wide bandgap power devices and features a negative temperature coefficient^{69,70}.”

Reference:

- [14] X. Zou, X. Zhang, X. Lu, C. W. Tang and K. M. Lau, "Fully Vertical GaN p-i-n Diodes Using GaN-on-Si Epilayers," in IEEE Electron Device Letters, vol. 37, no. 5, pp. 636-639, May 2016, doi: 10.1109/LED.2016.2548488.
- [15] Ruizhe Zhang and Yuhao Zhang, "Power device breakdown mechanism and characterization: review and perspective," Jpn. J. Appl. Phys. 62 SC0806, 2023.
- [16] A. O. Konstantinov, N. Nordell, Q. Wahab, U. Lindefelt; Temperature dependence of avalanche breakdown for epitaxial diodes in 4H silicon carbide. Appl. Phys. Lett. 28 September 1998; 73 (13): 1850–1852.
- [17] Z. Zhang et al., "Trap-Induced Leakage Current Increase in β -Ga₂O₃ Schottky Barrier Diodes Under 473-MeV Kr Ion Irradiation," in IEEE Transactions on Nuclear Science, vol. 72, no. 11, pp. 3543-3550, Nov. 2025, doi: 10.1109/TNS.2025.3620821.
- [18] Zhanbo Xia, Hareesh Chandrasekar, Wyatt Moore, Caiyu Wang, Aidan J. Lee, Joe McGlone, Nidhin Kurian Kalarickal, Aaron Arehart, Steven Ringel, Fengyuan Yang, Siddharth Rajan; Metal/BaTiO₃/ β -Ga₂O₃ dielectric heterojunction diode with 5.7 MV/cm breakdown field. Appl. Phys. Lett. 16 December 2019; 115 (25): 252104.
- [19] Y. Qin et al., "10-kV Ga₂O₃ Charge-Balance Schottky Rectifier Operational at 200 °C," in IEEE Electron Device Letters, vol. 44, no. 8, pp. 1268-1271, Aug. 2023, doi: 10.1109/LED.2023.3287887.
- [20] J. Liu et al., "Trap-Mediated Avalanche in Large-Area 1.2 kV Vertical GaN p-n Diodes," in IEEE Electron Device Letters, vol. 41, no. 9, pp. 1328-1331, Sept. 2020, doi: 10.1109/LED.2020.3010784.

Comment (6): In page 6, comes some comments on Fig. 3. In the caption it is written, "the insets show the optical images of the two prototyped submodules, where encapsulation is omitted to more clearly reveal the package structure.". That's a good idea.

Moreover in Supplementary Section S5 – Package Process, in the encapsulation section, it is written, "A 3D-printed polymer housing was glued to the DBC surface using a slow-curing epoxy resin (Bob Smith Industries). A two-part silicone gel (Wacker 612 A/B) was mixed, vacuum degassed to eliminate air bubbles". Unfortunately, there is no presentation of the housing. The polymer structure, as well, silicone gel are some limitations for an industrial product at 250 °C peak temperature. At least some comments or explanations must be given. For instance such housing is needed to protect the module against moisture and other environmental duties.

Response: We thank the reviewer for raising these important questions regarding the housing and silicone-gel encapsulation materials. As correctly noted by the reviewer, to better show the internal package structures, the optical images in the inset of Fig. 3a show sub-modules without encapsulation. In addition, all high-temperature forward and reverse I - V characterization were performed on devices before the final housing or silicone gel, to avoid the temperature limitations of these materials during the long characterizations at 250-300 °C.

The final sub-module uses a 3D-printed engineering-grade nylon housing (~150 °C thermal rating). The reviewer is correct that such polymer is not suitable for long-term operations at 250 °C. However, in this submodule, the housing is located far above the active device region. It mainly serves as a mechanical support and fixture to form a cavity for gel filling. It is not in direct thermal contact with the Ga₂O₃ chip, the BTO high-k interface, or the sintered-Ag layer. Thus, the housing does not experience the high junction-temperature during transient power events, particularly in pulsed power applications.

The encapsulant used in the sub-module is Wacker SilGel 612 A/B, a two-component silicone gel with high dielectric strength, very low modulus, long-term reliable temperature up to ~200 °C and excellent moisture barrier property. In the sub-module, silicone gel is used to provide isolation and anti-vibration protection and prevent moisture or ionic contamination. Importantly, the silicone gel is not placed in the high heat-flux region.

Because pulse power is highly transient (microseconds-milliseconds), a strong temperature gradient is present, and the encapsulant region remains far below 200 °C. The junction temperature during transient power may momentarily reach 250-300 °C. But the external package temperature remains much lower because of the extremely short pulse width and the low thermal diffusivity between junction and housing. Therefore, the over 200 °C capability demonstrated in Fig. 2 represents an intrinsic property of the Ga₂O₃ device itself and the JSC stack, not of the external housing or silicone gel.

We agree with the reviewer that for industrial-grade 250 °C modules (particularly for applications with longer pulses or even conventional switching applications), more robust encapsulation materials are needed. As part of our continuing development plan, we will migrate to high-temperature housings (e.g., PEI/Ultem, PEEK and Metal frame structures) for maximum thermal stability. For high-temperature encapsulants, ceramic-filled silicone gels and glass-based encapsulants are also in the further research, which can ensure true high-temperature reliability for industrial implementation.

Corresponding change in manuscript: Yes

Location of Change: The revisions have been incorporated into the last paragraph of the “**Supplementary Section S5 – Package Process**” section (Page 7) in the revised Supplementary Information.

Extended text:

“A 3D-printed engineering-grade nylon housing was glued to the DBC surface using a slow-curing epoxy resin (Bob Smith Industries). The housing mainly serves as a mechanical support and fixture to form a cavity for gel filling. A two-part silicone gel (Wacker 612 A/B) was mixed, vacuum-degassed to eliminate air bubbles, and poured into the housing to fully encapsulate the device and interconnects. The silicone gel is used to provide isolation and anti-vibration protection and prevent moisture or ionic contamination. The assembled modules were then cured at 150 °C for 30 minutes on a hotplate to complete the packaging process. The junction temperature during transient power may momentarily reach 250-300 °C. But the external package temperature remains much lower because of the extremely short pulse width and the low thermal diffusivity

between junction and housing. Therefore, the over 200 °C operation capability demonstrated in Fig. 2 reflects an intrinsic property of the Ga₂O₃ device itself and the JSC stack, not of the external housing or silicone gel.”

Comment (7): In Supplementary Section S7 - Temperature-dependent Breakdown Characterization of SiC SBD, it is shown the temperature limitation of the commercial SiC SBD (ROHM, S6304). The comparison is not fair because some research papers show better performance even for high voltage devices [1].

References

[1] KIMOTO, Tsunenobu, YAMADA, Kyosuke, NIWA, Hiroki, *et al.* Promise and challenges of high-voltage SiC bipolar power devices. *Energies*, 2016, vol. 9, no 11, p. 908.

Response: We appreciate the reviewer’s comment regarding the temperature limitation of the commercial SiC SBD (ROHM, S6304) in Supplementary Section S7 and the suggestion to compare with SiC devices reported in above reference cited by the review. Our intention in Supplementary Section S7 was not to claim an absolute breakdown limit of SiC, but to provide a practical benchmark against a commercially available 1.2-kV-class device with comparable drift-layer thickness and voltage rating to our Ga₂O₃ diode.

The paper cited by the reviewer (Kimoto et al., *Energies*, 2016, vol. 9, no 11, p. 908) is an excellent and comprehensive review on high-voltage SiC bipolar power devices, focusing on PiN structures or merged PiN-Schottky (MPS) structures for ultra-high-voltage (>10 kV) applications. In these devices, the drift layer is typically very thick (50-100 μm) and engineered with strong conductivity modulation (i.e., strongly bipolar), target breakdown voltage is over 10 kV.

In contrast, our work targets Ga₂O₃ diodes with very fast switching (i.e., ‘unipolar’) (note that although a bipolar NiO/Ga₂O₃ heterojunction is used, due to the very short hole lifetime in Ga₂O₃, the diode is nearly unipolar as proofed by the minimal reverse recovery), where the closest practical SiC counterpart in real applications is a 1.2-kV SiC SBD/JBS diode, rather than a >10 kV SiC PiN diode. Bipolar SiC PiN diodes are rarely used in standard 1.2-kV power conversion stages because of large forward voltage and significant reverse recovery loss due to minority carrier storage. Therefore, the cited reference by the review represents the ultimate potential of ultra-high-voltage SiC bipolar devices, but not the typical unipolar device class directly comparable to our Ga₂O₃ heterojunction diode in this work.

We intentionally chose a commercial 1.2-kV SiC SBD (ROHM S6304) with a close voltage rating and comparable drift-layer thickness. Although the ROHM S6304 is marketed as a “SiC SBD,” the bare die we obtained has a junction barrier Schottky (JBS) structure, which is in fact standard for most modern “SiC SBDs.” The device combines Schottky regions with buried p⁺ grids to suppress leakage. Under reverse bias, especially at high temperature, leakage is still predominantly governed by the metal/SiC Schottky barrier, leading to a pronounced increase of leakage with temperature. This behavior, rapidly increasing reverse leakage current and practical breakdown limitation at elevated temperature, is well reported for SiC Schottky/JBS diodes in the literature Ref. 21 and Ref. 22^{21,22}. While our Ga₂O₃ heterojunction diode maintains lower leakage and more robust high-temperature blocking under the same bias and test setup.

Corresponding change in manuscript: Yes

Location of Change: The revisions have been incorporated into the first paragraph of the “**Supplementary Section S7**” section (Page 10) in the revised Supplementary Information.

Expanded text:

“The SiC reference device used in Supplementary Section S7 is a commercial 1.2-kV SiC junction barrier Schottky (JBS) diode with a drift layer thickness comparable to our Ga₂O₃ device. Note that it is used to represent a typical industrial SiC SBD in the close voltage class, rather than the ultimate capability of high-voltage SiC bipolar devices that has been reported to be able to operate at very high temperatures¹⁵. The device combines Schottky regions with buried p⁺ grids to suppress leakage. Under reverse bias, especially at high temperature, leakage is still predominantly governed by the metal/SiC Schottky barrier, leading to a pronounced increase of leakage with temperature.”

Reference:

[21] R. Radhakrishnan, T. Witt and R. Woodin, "Temperature dependent design of Silicon Carbide Schottky diodes," 2014 IEEE Workshop on Wide Bandgap Power Devices and Applications, Knoxville, TN, USA, 2014, pp. 151-154, doi: 10.1109/WiPDA.2014.6964644.

[22] P. Brosselard et al., "Bipolar Conduction Impact on Electrical Characteristics and Reliability of 1.2- and 3.5-kV 4H-SiC JBS Diodes," in IEEE Transactions on Electron Devices, vol. 55, no. 8, pp. 1847-1856, Aug. 2008, doi: 10.1109/TED.2008.926636.

Comment (8): In Supplementary Section S9 - ANSYS Workbench simulation and calibration with experimental,

Figure S9-2a is confusing because a similar but different circuit is presented in Fig. 4a and the associated optical pictures are the same.

Some explanation must be given, at least for Fig. 4a, in the main text, where the SiC MOSFETs role must be given and the references of the devices are needed. Some explanations exist in Supplementary Section S11 – Experimental circuit testing setup for short-pulse DPT but the MOSFETs references are not given. Moreover, this also contributes to the idea that SiC MOSFETs can replace Ga₂O₃ diodes in a pulsed power application. Further explanation is needed.

Response: Thanks for the suggestion. The associated optical pictures in Figure S9-2a and Fig. 4a are different in all aspects, including PCB layout, device numbers, and connection configurations. It should be clarified that the dynamic switching test and the surge current test share the same physical circuit board with different connection configurations to achieve the target circuit functions (as seen in Fig. 5b and Supplementary Figure S9-2b), whereas the megawatt pulsed power test utilizes a completely separate, customized designed circuit board to represent the unique operation profiles in pulsed power applications (as seen in Fig. 4b). Conventional switching tests (e.g., double pulse test) primarily evaluate device behavior under moderate voltage and current levels, whereas pulsed-power systems deliver much larger energy within ultra-short and tunable pulse widths, requiring a highly compact circuit topology with minimized loop inductance and

extremely low parasitic resistance to support megawatt-level instantaneous power.

Fig. R9 illustrates the circuit schematic and PCB layout for the multi-functional board used in both the dynamic switching (Supplementary Figure S9-2b) and surge current tests (Fig. 5b). This board integrates two switches (S_1 and S_2), DC-link capacitors (C_{DC}), the DC power supply (V_{DC}), and the DUT. The circuit topology is reconfigured via specific connection nodes: for the surge current test, Node 1 (N1) is connected to the surge inductor (L_{surge}) and Node 2 (N2) is left open, reproducing the circuit in Fig. 5a in the manuscript. For the dynamic switching test, N1 is shorted while the switching inductor (L_{switch}) is connected to N2, reproducing the circuit in Supplementary Figure S9-2a. Fig. R9b highlights these connecting nodes to demonstrate how this design strategy shares key components to significantly reduce design cost.

Fig. R9. (a) Circuit schematic and (b) PCB layout of the multi-functional board used for both dynamic switching test and surge current test.

In contrast, the megawatt pulse power test board is different, and we have provided its specific PCB layout in Fig. R10 for reference to clearly distinguish it from the multi-functional board. The detailed circuit operation has been presented in Supplementary Section S11.

Fig. R10. PCB layout of the dedicated pulse power circuit board.

In Fig. 4a, the part number of the SiC MOSFETs used in megawatt pulsed power switching circuit is C3M0016120D from Wolfspeed, the rated voltage is 1200 V and the typical on-resistance (R_{on}) is 16 m Ω . These devices were chosen solely because they offer very low R_{on} and fast switching at 1.2-kV rating, enabling the generation of high-current, fast-rise-time pulsers required to stress our Ga₂O₃ diode module. Here we use four switching branches, each consisting of five SiC MOSFETs in parallel. The MOSFETs' only purpose is to control short-pulse, high-voltage, and high-current waveforms. These pulses are then delivered to the Ga₂O₃ diode module, which is the only rectifying element in the experiment. The reason for using many MOSFETs in parallel is to minimize the effective on-resistance of the switch array, achieve >1000 A pulsed current capability and reduce stress on individual MOSFETs during fast pulses.

As discussed in our **Response (1)** earlier, power diodes are mandatory components in pulsed-power circuits. In practical pulsed systems, such as Marx generators, pulse-forming networks, inductive storage, capacitive discharge systems, and high-current pulse chargers, diodes are essential for directing energy flow, isolating stages, clamping voltage transients, providing freewheeling paths during fast current interruptions and ensuring safe dead-time operation in converters. Although SiC MOSFET has intrinsic body diode, it suffers from bipolar degradation and considerable reverse recovery, associated with basal-plane dislocation expansion under forward conduction. The SiC MOSFET body diode is generally not used as a freewheeling or pulsed conduction diode in high-reliability power systems. Ga₂O₃ diodes exhibit several properties that make them fundamentally more suitable than SiC MOSFET body diodes in pulsed-power operation. The Ga₂O₃ diode maintains stable high-temperature characteristics up to 250 °C, high-power capacity density, superior surge current robustness, and fundamentally low- T_j rise due to Ga₂O₃'s higher volumetric heat capacity. In summary, based on the different functionalities of SiC MOSFETs and Ga₂O₃ diodes in pulsed power circuits as well as the fundamental enabling capabilities of Ga₂O₃ devices illustrated in this project, Ga₂O₃ diodes cannot be replaced by SiC MOSFETs in a pulsed power application.

Corresponding change in manuscript: Yes

Location of Change: The revisions have been incorporated into the 1st paragraph of the “**Circuit demonstration of megawatt pulsed-power switching**” section (Page 8) in the revised manuscript, and the 1st paragraph of the “**Supplementary Section S11**” section (Page 17) in the revised Supplementary Information.

Revised text:

- (1) “The DPT circuit consists of power SiC MOSFETs (C3M0016120D), an inductor, and the diode under test (DUT). The inductor is first pre-charged to a target current value, acting as a current source. The SiC MOSFETs are used to control short-pulse, high-voltage, and high-current waveforms with an adjustable pulse width, these pulses are then delivered to the Ga₂O₃ DUT for evaluating its turn-on and turn-off characteristics.”
- (2) “The part number of the SiC MOSFETs is C3M0016120D from Wolfspeed, the rated voltage is 1200 V and the typical on-resistance (R_{on}) is 16 mΩ.”

Comment (9): In figure S14, the color scale is not readable. Please correct that.

Response: We thank the reviewer for pointing this out. As shown in Fig. R11, the color scale in Fig. S14 has been enlarged and reformatted to improve readability. The updated figure has been included in the revised supplementary document.

Fig.R11. Simulated temperature distribution in the Ga₂O₃ sub-module and the hypothetical SiC submodule in the surge current test under different pulse widths.

Corresponding change in manuscript: Yes

Location of Change: The revisions have been incorporated into the Figure S14 of the “**Supplementary Section S14**” section (Page 22) in the revised Supplementary Information.

Comment (10): Fig. 3 | Thermal impedance characterization and time-resolved maximum power capacity derivation.

In Fig. 3e and Fig. 3f, it is clearly shown that the Ga₂O₃ module yields a better Z_{th} or maximum transient power capacity density, respectively. However, this is not entirely fair because the SiC diode uses a TO220 package and has a low current rating, 19 A. It is even possible to deduce that, to better respect the maximum power density need in a pulse power application, it is sufficient to choose a more recent diode with a higher current rating. So at least some additional explanations must be given.

Response: We thank the reviewer for the thoughtful comments regarding Fig. 3e and Fig. 3f. Our intention was to compare the intrinsic transient thermal behavior of various devices on a per-area basis, which is the physically meaningful metric for pulse-power maximum transient power capacity density. Moreover, from the discussion in this paper, in pulsed power applications, the shorter the pulse width, the less dependence of device maximum power capacity density on packages (as the heat diffusion is limited in the pulse transient and does not reach the package case); instead, it has a heavier dependence on material's inherent heat capacity and device's maximum operation temperatures.

Commercial SiC diodes typically use TO-220 packages, and higher current ratings imply proportionally larger die area. We agree with the reviewer that previous TO-220-packaged SiC diode (IDH02G120C5, Infineon) with small die area has a relatively small current rating. For transient maximum power capacity density analysis, we normalized the maximum allowable transient power by die area, because power capacity density (W/mm²) is the relevant figure of merit for pulse-power applications.

Following the reviewer's suggestion, we select a higher-current rated commercial SiC diode (IDWD20G120C5, Infineon) and performed a new comparison. The selected device exhibits a high continuous forward current of $I_F = 61$ A and a non-repetitive surge forward current of $I_{FSM} = 190$ A. This diode's thermal resistance is essentially identical to that of our Ga₂O₃ module, enabling a fair comparison, but this diode presents larger die area. As shown in Figure R12(a) and (b), The Ga₂O₃ module still demonstrates significantly higher maximum transient power capacity density across the entire time constant window. Even if SiC diodes are scaled to larger die areas, their transient maximum power capacity density remains fundamentally limited by SiC material and device properties. We have added Fig. R12a and Fig. R12b in the Supplementary Information to detail these analyses on high-current rated SiC diode.

Figure R12. (a) Derived transient thermal impedance of the Ga_2O_3 submodule and packaged SiC diode (IDWD20G120C5) and (b) extracted maximum transient power capacity density across various time scales.

Corresponding change in manuscript: Yes

Location of Change: The Fig. R12a-b and related explanation have been incorporated into the “**Supplementary Section S10 –Power capacity modeling**” section (Page 15) in the revised Supplementary Information.

Expanded text:

“Finally, we discuss the impact of packages and device selections on the above comparative analysis. First, from the later discussion in Supplementary Section S14, in pulsed power applications, the shorter the pulse width, the device maximum power capacity density is expected to be less dependent on packages but dominantly determined by material’s inherent heat capacity and device’s maximum operation temperatures. Thus, the TO package of SiC diode and the JSC package of our Ga_2O_3 diode will present relatively minimal difference for the thermal transients at short pulse duration. Second, to investigate the analysis’ dependence on device selection, we select a higher-current rated commercial SiC diode (IDWD20G120C5, Infineon) for a new comparison, which exhibits a high continuous forward current of $I_F= 61$ A and a non-repetitive surge forward current of $I_{FSM}= 190$ A. This thermal resistance is nearly identical to that of our Ga_2O_3 module, enabling a fair comparison. As shown in Figure S10-2(a) and (b), the Ga_2O_3 module still demonstrates significantly higher maximum transient power capacity density across the entire time window. This suggests, even if SiC diodes are scaled to larger die areas, their transient maximum power capacity density remains fundamentally limited by SiC material properties.”

Comment (11): Fig. 4 | Demonstration of megawatt pulsed power switching in practical power converters. Concerning this figure, d) simulated time-resolved T_j evolution of the Ga_2O_3 sub-module with high- κ interface, but the associate figure shows two curves. The read one (257 A) has a written label “breakdown” without any explanation. Some explanations or corrections must be given.

Response: We thank the reviewer for pointing out that Fig. 4d contains two simulated curves, and that the label “breakdown” associated with the 257-A condition was not adequately explained in the original submission. We fully agree and have added explicit discussion in the main text. Fig. 4d illustrates the simulated time-resolved junction-temperature evolution of our Ga₂O₃ sub-module during forward conduction followed by a forced reverse blocking transition. When the switching current increases to 257 A, the device undergoes the following sequence: During forward conduction (10-15 μs window), power dissipation causes the junction temperature to rise rapidly, and the simulated peak junction temperature reaches over 300 °C. Immediately after switching to high reverse bias, at ~15 μs, the conduction is cut off and the diode is forced to block over 1 kV. However, at this instant the device is already near the high-temperature limit observed in static measurements (see Fig. 2d in the main text). At this instant, the combination of high T_j and high reverse electric-field stress causes irreversible failure. Thus, when the device switches into the reverse-blocking state at over 300 °C, it can no longer sustain the reverse blocking requirement, leading to thermal-induced breakdown. After reverse blocking capability collapses, the diode effectively behaves as a short circuit, resulting in sudden conduction of large reverse current, further power dissipation, and continued rise in simulated temperature beyond 15 μs.

Corresponding change in manuscript: Yes

Location of Change: The revisions have been incorporated into the 1st paragraph of the “**Circuit demonstration of megawatt pulsed-power switching**” section (Page 8) in the revised manuscript.

Expanded text:

“... When the blocking voltage degrades below the bus voltage, breakdown occurs.”

Comment (12): Fig. 5 | Surge current capability of Ga₂O₃ sub-module in conduction-dominant pulsed power applications.

In the caption, it is written, “... tested with an increased surge energy at a fixed pulse width of 3.6 μs. The critical peak surge current is over 800 A.” For the top curve, it is written “device degradation”, but no explanation is given. Please correct this.

Response: We thank the reviewer for pointing out the lack of explanation regarding the “device degradation” label in Fig. 5. We clarify that this label refers to measurable degradation of reverse-blocking capability after a surge-current event, rather than immediate device destruction during the pulse itself. After each surge current test, forward and reverse I - V characteristics as well as breakdown voltage was measured to examine any possible degradation. When the post-surge reverse-blocking capability remained unchanged, the surge was deemed safe. When we observed noticeable increase in leakage current, or a measurable drop of breakdown voltage, we defined this condition as device degradation. Thus, degradation refers to a post-surge deterioration of device blocking performance. As shown in Fig. 5c in the main test, the Ga₂O₃ device sustains a peak surge current of 856 A, when we further increase the current, a noticeable degradation in breakdown was observed. Therefore, 856 A is determined to be the maximum acceptable surge-current level before noticeable degradation occurs. Beyond this level, the device does not catastrophically fail, but its reverse-

blocking margin begins to deteriorate. The degradation mechanism is consistent with that under the high-temperature pulse circuit tests that involves both on-state current conduction and off-state high-voltage blocking, which has been discussed in detail in the last response.

Corresponding change in manuscript: Yes

Location of Change: The revisions have been incorporated into the 3rd paragraph of the “**Short-pulse surge current capability**” section (Page 9) in the revised manuscript.

Expanded text:

“Post-test static characterization confirms the device integrity until the last 856 A surge current test, after which the device exhibits a measurable deterioration in its reverse-blocking capability.”

Comment (13): Concerning the “Methods section”, it's rather confusing because the purpose of this section, after the conclusion, is unclear. Probably, a section before the conclusion with a small introduction to explain the purpose of the section will be clearer.

Response: We thank the reviewer for the suggestion regarding the placement and clarity of the “Methods” section. We fully understand that the structure may appear unconventional compared to other journal formats (e.g., IEEE). However, we would like to clarify that the current position and formatting of the Methods section strictly follow Nature Communications manuscript preparation guidelines. This structure is required for Nature Communications articles and adheres to the format recommended in their official author instruction. The major purpose of placing Methods after the main sections in Nature-series papers is to present the major results precisely in the main sections, which could be of interest to readers in a broad community, while leaving the technical details in the Methods and Supplementary Information to inform readers in a narrower and more specific field.

Corresponding change in manuscript: No

Comment (14): The conclusion is very positive for the performance of Ga₂O₃ modules for “pulsed power switching at 1000 A, 1000 V”. However, an analysis of the road map to industrial products should be very interesting for most readers. For instance in [2] it is written (by some of the same authors), “We envision the following immediate research gaps that need to be addressed for advancing Ga₂O₃ into industrial power electronics applications”.

Probably, a similar analysis of pulse power applications will be very useful.

[2] QIN, Yuan, WANG, Zhengpeng, SASAKI, Kohei, *et al.* Recent progress of Ga₂O₃ power technology: large-area devices, packaging and applications. *Japanese Journal of Applied Physics*, 2023, vol. 62, no SF, p. SF0801.

Response: We thank the reviewer for encouraging a forward-looking discussion on the industrial roadmap of Ga₂O₃ devices for pulsed-power applications. The demonstrated 1000 A / 1000 V pulsed-power switching performance highlights the intrinsic advantages of Ga₂O₃ for ultra-high-power-density, short-duty-cycle operations and reaches a power regime directly relevant to practical pulsed-power systems, indicating clear

potential for near-term industrial adoption. To further accelerate this transition, continued scaling of power capacity is highly desirable. At present, power devices operating at multi-kiloamp and multi-kilovolt levels are predominantly based on Si bipolar technologies (e.g., IGBTs, IGCTs, and thyristors), which inherently suffer from slow switching speeds and large reverse-recovery losses. The availability of a fast-switching, compact UWBG module at comparable power levels would therefore offer a compelling performance advantage.

To enable such power scaling in Ga₂O₃ devices and modules, advances in current rating, voltage rating, and operating temperature are required. From a material perspective, this necessitates improved wafer uniformity with reduced defect densities to support larger die areas and higher per-die current, as well as thicker, low-doped epitaxial layers to sustain higher blocking voltages. At the device and module levels, further optimization of heterogeneous interfaces in both devices and packaging will be essential to enable reliable high-voltage operation under the combined high temperature and electric-field conditions. Following these considerations, we have added a corresponding forward-looking paragraph to the Conclusion of the revised manuscript.

Corresponding change in manuscript: Yes

Location of Change: The revisions have been incorporated into the “**Conclusions**” section (Page 11) in the revised manuscript.

Expanded text:

“Looking forward, further upscaling of Ga₂O₃ devices and modules toward higher power capacity will require concurrent advances in current and voltage ratings, as well as high-voltage reliability at temperatures beyond the 250 °C demonstrated here. From a material perspective, this calls for the development of larger-diameter Ga₂O₃ wafers with improved uniformity and reduced defect density, together with thicker and low-doped epitaxial layers to support higher voltages. At the device and module levels, continued optimization of edge terminations and heterogeneous interfaces in both devices and packaging will be essential to alleviate degradation mechanisms that are accelerated by the combined effects of high electric field and elevated temperature. Beyond diode demonstrations, the realization of Ga₂O₃ transistors that leverage the device-package co-design principles identified in this work will be critical for enabling fully integrated, all-Ga₂O₃ pulsed-power systems.”

Comment (15): Conclusion: the paper needs major revisions.

Response: We thank the reviewer for the overall evaluation. We have carefully revised the manuscript according to all comments, we believe the revised version has been significantly improved.

2. Reply to the 2nd reviewer's comments

Reviewer #2 (Remarks to the Author):

The proposed methodology is novel and well-executed, integrating device-level dielectric engineering with packaging-level electrothermal co-optimization. The incorporation of a high-k BaTiO₃ interface for field flattening, combined with junction-side cooling and optimized die attach design, effectively address high electric field and thermal management challenges. The authors also demonstrate a deep understanding of Ga₂O₃'s intrinsic properties, particularly its high volumetric heat capacity and superior high temperature stability, and skillfully design both the device and packaging to exploit these advantages for the high-pulse power applications. The module level results are encouraging and support the technical soundness of the concept. Overall, this thoughtful UWBG co-design represents a promising step toward the practical deployment of Ga₂O₃ based modules, particularly for high pulse energy use.

Response: We sincerely thank the reviewer for the positive and encouraging comments. We are glad that the methodology and experimental results are well recognized, and we appreciate the reviewer's acknowledgment of our approach and findings. Thank you for the supportive feedback.

Comment (1): It would be beneficial to add some comments on the dipole properties of BTO and why this material is the superior choice to others that may also possess such properties. I.e, why BTO?

Response: We thank the reviewer for the suggestion and have revised the manuscript accordingly. Our motivation for selecting BTO is primarily rooted in its high dielectric permittivity and the resulting electric-field modulation functionality under reverse bias. When the Ga₂O₃ drift region becomes depleted, fixed positive charge exists in the depletion region. Owing to the large permittivity contrast between BTO and Ga₂O₃, the external electric field induces a polarization gradient inside the BTO layer, creating net negative bound polarization charge at the BTO/Ga₂O₃ interface and a corresponding positive charge beneath the metal/BTO interface. As shown in Fig. R13, the negative bound charge effectively compensates the positive charge in the depletion region, reducing the local electric-field peak and flattening the field distribution. This permittivity-induced charge balancing is similar to conventional superjunction compensation, where equal positive and negative charges flatten the electric field and decouple breakdown voltage from sheet charge density.

In theory, this polarization superjunction functionality will hold for all high-permittivity dielectrics. Here, BTO is selected not only for its high permittivity (measured to be 155 from our *C-V* characterization, as shown in Supplementary Information S3), but also for its fabrication simplicity, low leakage current, and proven compatibility with Ga₂O₃ to form high-quality heterojunctions^{23,24}. In addition, BTO has been successfully applied in both Ga₂O₃ and GaN based power devices, demonstrating superior capability in improving surface electric-field distribution and enhancing breakdown voltage²⁵⁻²⁷. Therefore, BTO represents an optimal dielectric choice for this work.

Fig. R13. Illustration of the polarization dipoles introduced by the high- κ dielectrics in the metal/BaTiO₃/Ga₂O₃ structure

Corresponding change in manuscript: Yes

Location of Change: The revisions have been incorporated into the 3rd paragraph of the “**Device-package electrothermal co-optimization**” section (Page 6) in the revised manuscript.

Expanded text:

“As illustrated in Fig. 1f, owing to the large permittivity contrast between BTO and Ga₂O₃, the external electric field induces a polarization gradient inside the BTO layer, creating net negative bound polarization charge at the BTO/Ga₂O₃ interface and a corresponding positive charge beneath the metal/BTO interface^{64,65}. This polarization dipole can shield the electric field produced by charges in the metal and enable a nearly-zero net charge to flatten the crowded electric field – similar to the legacy superjunction design in multidimensional power devices⁶⁶.”

Reference:

[23] S. Roy, A. Bhattacharyya, P. Ranga, H. Splawn, J. Leach and S. Krishnamoorthy, "High-k Oxide Field-Plated Vertical (001) β -Ga₂O₃ Schottky Barrier Diode With Baliga's Figure of Merit Over 1 GW/cm²," in IEEE Electron Device Letters, vol. 42, no. 8, pp. 1140-1143, Aug. 2021, doi: 10.1109/LED.2021.3089945.

[24] Xia, Z. *et al.* Metal/BaTiO₃/ β -Ga₂O₃ dielectric heterojunction diode with 5.7 MV/cm breakdown field. *Applied Physics Letters* **115** (2019).

[25] Roy, S., Bhattacharyya, A., Peterson, C. & Krishnamoorthy, S. β -Ga₂O₃ lateral high-permittivity dielectric superjunction Schottky barrier diode with 1.34 GW/cm² power figure of merit. IEEE Electron Device Lett. 43, 2037-2040 (2022).

[26] Kyle J. Liddy, Weisong Wang, Stefan Nikodemski, Chris Chae, Kevin D. Leedy, Jean-Pierre Bega, Nolan S. Hendricks, Elizabeth A. Sowers, Ahmad E. Islam, Jinwoo Hwang, Siddharth Rajan, Andrew J. Green; Ultra-high permittivity BaTiO₃ ($\epsilon = 230$) on Al₂O₃/AlGaIn/GaN MISHEMTs for field-management in high-voltage RF applications. APL Electronic Devices 1 March 2025; 1 (1): 016112.

[27] Mohammad Wahidur Rahman, Nidhin Kurian Kalarickal, Hyunsoo Lee, Towhidur Razzak, Siddharth Rajan; Integration of high permittivity BaTiO₃ with AlGaIn/GaN for near-theoretical breakdown field kV-class transistors. Appl. Phys. Lett. 8 November 2021; 119 (19): 193501.

Comment (2): Will the high-k BaTiO₃ interface layer functionally replace the conventional SiO₂/SiN₄ surface passivation on chips, or is it intended to be added on top of the conventional passivation to act as an additional field-modulation layer? If it is a replacement, please comment on its insulating reliability and long-term stability? How will it stand up to humidity, a known issue with BaTiO₃? I.e., will the field dampening properties be sacrificed? If so, by how much?

Given the CTE mismatch between BaTiO₃ and Ga₂O₃, how robust is the interface during repeated thermal cycling and high-field pulsed operation? Please comment on potential interfacial defects, charge trapping/polarization drift, breakdown-voltage stability, and any signs of delamination or micro-cracking.

Response: We sincerely thank the reviewer for this insightful and constructive comment. In the present work, the BaTiO₃ layer is not intended to replace the conventional chip-level surface passivation (e.g., SiO₂/SiN_x). Instead, it acts as an additional high-k field-modulation/interface layer inside the junction-side cooling (JSC) stack that interfaces between the chip surface (i.e., passivation layer) and die attach metals (e.g., sintered Ag). The reviewer is correct that humidity sensitivity is a known issue for exposed BTO material. In our package and module, the external package surface is further protected by the module encapsulation (housing and silicone gel), so the BTO is not directly exposed to the ambient environment. Silicone gel is injected to form a continuous dielectric barrier covering the device surface and sidewalls. The gel is then vacuum-degassed to eliminate air bubbles and voids, which are known to act as local moisture traps and accelerate dielectric degradation. After degassing, the encapsulant is thermally baked to remove residual moisture content, ensuring that the internal cavity remains dry. As a result, BTO is not subjected to open-environment humidity exposure, alleviating one of the main degradation concerns typically associated with perovskite-based dielectrics.

In addition, the BTO layer is deposited as a thin sputtered film in this work, rather than as a bulk ceramic. Thin-film BTO exhibits significantly lower internal porosity, reduced grain-boundary conduction pathways, and reduced ionic migration compared to bulk sintered BaTiO₃ used in MLCCs, which partially mitigates humidity- and defect-induced failure modes that are widely reported in ceramic capacitors.

Table I. The list of key material CTE.

Material	CTE (10 ⁻⁶ /K)
Sintered Ag	~20
SiO ₂	~0.55
BaTiO ₃	10.8 - 17.5
Ga ₂ O ₃	3.77 (a), 7.8 (b), 6.34 (c)

We also appreciate the reviewer brings up the possible concern about the coefficient of thermal expansion (CTE) mismatch between BaTiO₃ and Ga₂O₃. Table I summarizes the CTE of Ga₂O₃, SiO₂, BaTiO₃, and sintered Ag. Owing to the large CTE mismatch among Ga₂O₃, SiO₂, and sintered Ag, the

Ga₂O₃/SiO₂/Ag stack in sub-module A experiences pronounced thermo-mechanical stress at the die edges. In contrast, replacing SiO₂ with BaTiO₃ improves the CTE match in the Ga₂O₃/BaTiO₃/Ag stack, thereby mitigating edge-induced stress concentration. This trend is confirmed by thermo-mechanical simulations in ANSYS. Fig. R14 shows the equivalent (von Mises) stress in the sintered-Ag layer for the two submodules (sub-module A with SiO₂ interface, sub-module C with BaTiO₃ interface) under identical boundary conditions (convection coefficient of 1000 W·m⁻²·K⁻¹ and 20 W power loss). The Ag edge stress in sub-module A reaches ~17 MPa, whereas it is reduced to ~12.5 MPa in sub-module C, evidencing the benefit of the better CTE matching provided by BaTiO₃.

Figure R14. Simulated equivalent (Von-Mises) stress in the sintered silver layers of (a) sub-module A with SiO₂ interface and (b) sub-module C with BaTiO₃ interface.

To evaluate long-term reliability under realistic thermal-electrical stress conditions, we performed a switching-based power cycling test, which is one of the widest used methods for industrial module qualification²⁸. Fig. R15a and Fig. R15b show the schematic and photo of switching-based power cycling circuit. Initially, S1 and S3 are turned on, generating a high current within the rating range of the DUT. This current flows through the loop formed by S1, S3, L, and the DUT. Next, S1 is turned off and S2 is turned on, forcing the current to freewheel through the loop consisting of S2, L, and the DUT. During this process, the freewheeling time is relatively long, gradually heating the DUT. By repeating this cycle, the DUT can be gradually heated to the target temperature within 60 seconds. The diode is then switched off, allowing it to cool down to room temperature. This heating-cooling cycle is repeated to achieve power cycling. Fig. R15c presents the injected power profile applied to device and case temperature profile. In the subsequent 120-s cooling period, a forced-air convection is applied to rapidly dissipate accumulated heat. Case temperature evolution is continuously monitored using an infrared thermal-imaging camera, and the peak case temperature reaches ~100 °C.

Figure R15. (a) Schematic and (b) photo of power cycling test circuit. (c) Injected power profile applied to device and case temperature profile. (d) Illustration of dynamic $I-V$ reconstruction from the switching current and voltage waveforms recorded in each switching cycle. (e) Extracted V_F during power cycling test for sub-module A with SiO_2 interface and sub-module C with BTO interface.

At the end of each cooling stage, we capture the current and voltage waveforms under the same excitation conditions and reconstruct a dynamic $I-V$ curve by extracting a series of (I, V) points, each taken at the same time instant (Fig. R15d). From the re-constructed dynamic $I-V$ characteristics, the forward-voltage drop (V_F) is extracted at a fixed current of 5 A. Fig. R15e shows that sub-module C exhibits minimal V_F shift over $\sim 10,000$ power-cycling cycles, indicating stable conduction characteristics. In addition, and we did not

observe measurable degradation in reverse blocking capability. In contrast, sub-module A exhibits serious V_F degradation after ~1,200 cycles, suggesting the possible formation of delamination, crack, and voids in the package interface. This contrast proves that submodule C with BTO interface demonstrates the high structural and electrical robustness under dynamic switching stress.

Corresponding change in manuscript: Yes

Location of Change: The revisions have been incorporated into the last paragraph of the “**Short-pulse surge current capability and Power cycling test**” section (Page 10) in the revised manuscript; a new section “Supplementary Section S15-Material CTE and power cycling test” is added to the Supplementary Information (Page 24).

Expanded text in the manuscript:

“Finally, we evaluate the reliability of the prototyped submodules. The BaTiO₃ layer is fully encapsulated by silicone gel, forming a continuous dielectric barrier that suppresses humidity-induced degradation commonly observed in bulk BaTiO₃ ceramics⁷⁶. In addition, the BaTiO₃ interface is found to improve thermo-mechanical reliability. Table S3 in **Supplementary Section S16** summarizes the CTE of relevant materials. Due to the large CTE mismatch, conventional Ga₂O₃/SiO₂/Sintered Ag stacks experience elevated thermo-mechanical stress near die edges. Replacing SiO₂ with BaTiO₃ significantly improves CTE matching. To validate this, we perform a switching-based power-cycling test, the widely-used method for qualifying power module reliability⁷⁷. As detailed in **Supplementary Section S16**, the sub-module C exhibits minimal parametric shift after 10,000 power-cycling cycles, whereas sub-module A exhibits pronounced forward voltage degradation after ~1,200 cycles.”

Reference:

[28] Y. Zhang et al., "Power Cycling Testing for Power Semiconductor Switches: Methods, Standards, Limitations, and Outlooks," in IEEE Transactions on Power Electronics, vol. 41, no. 1, pp. 849-869, Jan. 2026, doi: 10.1109/TPEL.2025.3595180.

Comment (3): BaTiO₃ is deposited by RF magnetron sputtering in an Ar/O₂ ambient. How compatible is this process with large-area fabrication and downstream metallization/packageing?

Response: We thank the reviewer for this question. The BaTiO₃ layer used in this work is deposited by RF magnetron sputtering in Ar/O₂ ambient, which is a well-established technique for large-area dielectric thin-film fabrication. The scalability primarily depends on the sputtering chamber configuration, including target diameter and substrate carrier size, rather than the material itself. Commercial sputtering systems already support 4-inch to 8-inch wafers using large-size targets. Therefore, extending BaTiO₃ deposition to large-area substrates is not a constraint. The process operates at relatively low substrate temperature, thus maintaining full compatibility with downstream metallization and packaging flows. In our experiments, the sputtered BaTiO₃ exhibited stable adhesion to subsequent metal deposition, as well as strong bonding to the

sintered Ag layer in the module assembly, without additional surface treatment. Therefore, the deposition process is compatible with large-area fabrication and does not impose limitations on subsequent process integration.

Corresponding change in manuscript: No

Comment (4): Can the high-k interface be extended to other WBG/UWBG devices (e.g., SiC, GaN, diamond)? If so, please outline material-specific adaptations, interface reliability under cycling, CTE matching, and process compatibility.

Response: We thank the reviewer for the insightful question. Yes, the high-k BaTiO₃ interface concept can be extended to other WBG/UWBG platforms, including SiC, GaN, and diamond, although material-specific adaptations will be required. In terms of process compatibility, RF magnetron sputtering of BaTiO₃ is a widely used and scalable thin-film approach. The deposition process can be directly adapted to SiC and GaN wafers, and thin-film perovskites have already been demonstrated on this semiconductor platforms²⁹⁻³². The scalability mainly depends on chamber configuration, rather than on material limitations. However, the film morphology and interfacial trap density may vary depending on the underlying substrate surface quality, which requires careful investigations. For thermal-mechanical robustness, CTE matching becomes material-dependent. For GaN, the mismatch relative to BaTiO₃ is moderate, so thin-film BaTiO₃ can be directly integrated without introducing considerable strain. For SiC and diamond, however, the CTE difference is significantly larger. Thus, BaTiO₃ would likely require thickness optimization, compliant transition layers, or stress-relaxation interlayers to avoid interfacial cracking under repetitive thermal cycling. Therefore, the high-k field-modulation concept is broadly transferrable to other WBG/UWBG systems, but film thickness, interface preparation, CTE matching strategy, and long-term reliability assessment will need to be tailored to each material system.

Corresponding change in manuscript: No

References:

- [29] Kyle J. Liddy, Weisong Wang, Stefan Nikodemski, Chris Chae, Kevin D. Leedy, Jean-Pierre Bega, Nolan S. Hendricks, Elizabeth A. Sowers, Ahmad E. Islam, Jinwoo Hwang, Siddharth Rajan, Andrew J. Green; Ultra-high permittivity BaTiO₃ ($\epsilon = 230$) on Al₂O₃/AlGaIn/GaN MISHEMTs for field-management in high-voltage RF applications. *APL Electronic Devices* 1 March 2025; 1 (1): 016112.
- [30] Mohammad Wahidur Rahman, Nidhin Kurian Kalarickal, Hyunsoo Lee, Towhidur Razzak, Siddharth Rajan; Integration of high permittivity BaTiO₃ with AlGaIn/GaN for near-theoretical breakdown field kV-class transistors. *Appl. Phys. Lett.* 8 November 2021; 119 (19): 193501.
- [31] Choi, JS., Lee, HW., Lee, TH. et al. Effects of post-deposition annealing on BaTiO₃/4H-SiC MOS capacitors using aerosol deposition method. *Appl. Phys. A* 130, 188 (2024).
- [32] Thirumaleshwara N Bhat et al. Polarization-induced interfacial coupling modulations in BaTiO₃/GaN heterojunction devices. 2017 *J. Phys. D: Appl. Phys.* 50 275101.

3. Reply to the 3rd reviewer's comments

Reviewer #3 (Remarks to the Author):

Response: We sincerely thank the reviewer and appreciate the effort contributed to this review. We also acknowledge and value the participation of Early Career Researchers in the evaluation process. Their constructive comments and feedback have helped us significantly improve the manuscript. Thank you again for your time and support.

4. Reply to the 4th reviewer's comments

Reviewer #4 (Remarks to the Author):

This is an excellent manuscript highlighting the performance of Ga₂O₃ UWBG power semiconductor devices, especially for pulsed power applications. The authors provided a comprehensive literature review, indicating the unique challenges and how Ga₂O₃ can offer potential solutions. The emphasis on the electro-thermal co-design approach is a new trend for all future power electronic systems. The experimental demonstration of the Mega Watt pulse power is also impressive, completed with device simulation, packaging design, electrical and thermal considerations. I truly enjoy reading this paper.

Response: We sincerely thank the reviewer for the very positive and encouraging comments. We are glad that the co-design approach, experimental demonstrations, and technical contributions were well recognized. Your supportive feedback is greatly appreciated and has motivated us to further refine and strengthen the work. Thank you again for your thoughtful review.

Comment (1): Perhaps one comment that may help further improve this paper is to discuss how the repetition rate (or duty cycle) has an effect on the thermal capacity of the proposed power module. Will the proposed pulse power application require any thermal exchanger to further improve the performance?

Response: We appreciate the reviewer's insightful comment. The repetition rate (or duty cycle) directly determines the average power dissipation of the module. For a given pulse energy, increasing the repetition rate increases the average heat load. To evaluate the impact of repetition rate (or duty cycle), we performed transient electro-thermal simulations using ANSYS, in which the Ga₂O₃ power sub-module is mounted on a backside heat sink and cooled by forced air (fan cooling), as shown in Fig. R16.

Figure R16. Ansys simulation setup of Ga₂O₃ power submodule with a backside heatsink and cooling fan.

To explicitly study the repetition-rate effect, the applied power dissipation was modeled as a square-wave pulse, with a peak power of 10 kW and pulse width of 5 μ s. Two repetition rates were considered: 1 kHz (duty cycle = 1/200) and 5 kHz (duty cycle = 1/40). The simulation setup schematic is shown in Fig. R17a. The corresponding transient junction-temperature (T_j) responses are compared in Fig. R17b. Owing to the large simulation data volume, only the first 5 ms of junction-temperature evolution is presented.

At 1 kHz, the average power dissipation is very small; thus, the average junction temperature barely increases, and the peak transient temperature rise is governed by the thermal capacitance of the device and package. When the repetition rate increases to 5 kHz, heat accumulation becomes pronounced because the generated heat cannot be fully dissipated between pulses. As a result, an elevated junction temperature develops in addition to the repetitive transient temperature peaks. While the transient peaks are still determined by the thermal capacitance of the device and package, the average junction temperature rise is strongly influenced by long-timescale heat removal, including junction-side cooling and the thermal conductivity of the packaging materials in our sub-module structure.

Figure R17. (a) Simulation setup schematic of applied power dissipation at different repetition rate. (b) Transient junction-temperature responses.

This average junction temperature rise can be effectively suppressed by more efficient external cooling. As shown in Fig. R18, under conventional forced-air cooling, a steady-state junction temperature of 44.8 °C is obtained under an average power of 5 W; whereas increasing the repetition rate leads to an average power of 25 W, resulting in a much higher steady-state junction temperature of 140.6 °C. By enhancing the thermal exchanger (e.g., stronger forced-air cooling), the steady-state junction temperature can be reduced to 111.3 °C. Further reduction is achievable by adopting more advanced cooling schemes, such as liquid cooling.

Overall, the simulations indicate the important impact of duty cycle on the thermal performance of the Ga₂O₃ module. For low duty cycles, thermal exchangers and external cooling play a relatively minor role,

and the power capacity is mainly determined by the intrinsic properties of the device and module. For higher duty cycle pulsed applications (and even approaching the conventional power switching applications), thermal exchangers and external cooling become increasingly important and can become new limitations for the power capacity; To fully exploit the intrinsically high power capability of the Ga₂O₃ module, advanced thermal exchanger designs and enhanced cooling strategies are therefore required in this scenario.

Figure R18. Transient time-resolved T_j simulation at various power and cooling conditions.

Corresponding change in manuscript: Yes

Location of Change: The revisions have been incorporated into the 5th paragraph of the “Circuit demonstration of megawatt pulsed-power switching” section (Page 10) in the revised manuscript. In addition, a new section “Supplementary Section S15-Transient junction-temperature simulation under varying duty cycles” has been added to the Supplementary Information (Page 23).

Expanded text in the manuscript:

“A similar consideration is also relevant for repetitive switching under varying duty cycles, or equivalently, different repetition rates. As shown in the analysis in **Supplementary Section S15**, at relatively high duty cycles or repetition rates, an average junction temperature rise develops in addition to the transient pulsed temperature peaks when the generated heat cannot be fully removed between pulses. This average junction temperature rise can be suppressed through improved external cooling. In this scenario, advanced external cooling is required to fully exploit the module’s intrinsic power capacity.”

5. Reply to the 5th reviewer's comments

Reviewer #5 (Remarks to the Author):

The paper demonstrates the first ultra-wide bandgap (UWBG) power module, which enhances the power capacity of UWBG devices by over three orders of magnitude and identifies pulsed power electronics as a novel application for Ga₂O₃ power devices. Considering the rapidly growing interest in UWBG power electronics and huge industrial investment in this domain, such a performance breakthrough is expected to have significant impact. This work also presents fundamental understanding by providing a high-κ device-package interface and highlighting the significance of heat capacity as a critical material attribute for pulsed power electronics, suitable to various different semiconductors. This work could attract widespread attention from this perspective. The manuscript is clear, well written and well structured. The data are comprehensive, covering device, packaging, and circuit-level verification, supported by theoretical analysis. To further improve the paper, the following technical points should be clarified:

Response: We sincerely thank the reviewer for the positive and encouraging comments and for recognizing the significance, impact, and clarity of this work.

Comment (1): For experimental circuit tests, is it possible to measure the power module under higher slew rate and switching frequency? The authors are also suggested to comment on the limiting factors for slew rate and frequency in this circuit test.

Response: We thank the reviewer for this insightful comment. The experimental circuit in this work is designed to emulate many typical pulsed-power switching applications that emphasize on delivering high peak current and high transient power within microsecond-scale pulses with a relatively small duty cycle. Such operating conditions are representative of a wide range of pulse-power systems. For example, in the Marx pulse generator for high-power pulse applications, the pulse widths range from 50 to 1000 ns with small duty cycle less than 1/100³³.

In the present experiment measurements, the minimum achievable pulse width is 2.5 μs, and a peak current slew rate (di/dt) of approximately 1.6 kA/μs is obtained. The achievable slew rate and pulse width in the experimental setup are primarily limited by the parasitic capacitance and inductance of the test circuit, as well as the switching speed of the SiC MOSFETs used to drive the module. The circuit was carefully optimized to minimize parasitics, and the achieved operating conditions already push the practical limits of the current test platform.

Regarding switching frequency, as discussed in the **Response of Reviewer #4**, in repetitive switching, switching frequency can impact the module's thermal performance. From the analysis in **Supplementary Section S15**, while the power capacity under short pulse widths and small duty cycles is mainly determined by the module's intrinsic properties, at relatively high duty cycles or frequencies, an average junction temperature rise develops in addition to the transient pulsed temperature peaks when the generated heat cannot be fully removed between pulses. This average junction temperature rise can be suppressed through

improved external cooling. In this scenario, advanced external cooling is therefore required to fully exploit the module's intrinsic power capacity.

We also understand that high dv/dt slew rate is also required in many other pulsed-power applications, such as nuclear and fusion systems, high-power particle accelerators, pulsed X-ray sources, and electromagnetic launchers. In these applications, pulse-forming networks, Marx generators, and fast solid-state switches generate sub-microsecond voltage transitions, leading to high dv/dt over 10 V/ns^{34,35}. At present, we do not observe any fundamental limitation that would prevent our module from operating under such high dv/dt slew rates. From a thermal perspective, this regime closely resembles a short-pulse operation, in which the temperature rise is dominated by thermal capacitance. Consequently, the Ga₂O₃ submodule is expected to sustain an even higher power capacity under these conditions. This expectation is further supported by the results in Fig. 5 of the manuscript, where the surge-current capability—and the corresponding power capacity assuming a fixed switching voltage—increases with decreasing pulse width. Experimental validation at such high dv/dt requires specialized test platforms that are currently unavailable to our group and will be pursued in future work through collaborations with researchers in plasma and nuclear sciences.

Corresponding change in manuscript: No

References:

- [33] C. Yao et al., "Parallel SiC MOSFETs Marx Pulse Generator Based on Magnetic Induction Current Balance," in IEEE Transactions on Plasma Science, vol. 52, no. 9, pp. 4250-4259, Sept. 2024, doi: 10.1109/TPS.2024.3369047.
- [34] K. Wen, L. Liang, Z. Zhang and X. Yan, "High Power and High Repetition Frequency Nanosecond Marx Generator Based on Avalanche Transistor With Triggering Acceleration Circuit," in IEEE Transactions on Power Electronics, vol. 40, no. 6, pp. 8153-8167, June 2025, doi: 10.1109/TPEL.2025.3534802.
- [35] M. -K. Nguyen, F. Zare and N. Ghasemi, "Switched-Capacitor-Based Nanosecond Pulse Generator Using SiC MOSFET," 2018 Australasian Universities Power Engineering Conference (AUPEC), Auckland, New Zealand, 2018, pp. 1-6, doi: 10.1109/AUPEC.2018.8757883.

Comment (2): Ga₂O₃ devices are known to be promising for high-voltage applications, with devices up to 10,000 V having been experimentally reported. Does the superior power capacity of Ga₂O₃ retain for higher-voltage devices? The authors may add supplemental simulations.

Response: We thank the reviewer for this insightful comment. Recently, Ga₂O₃ devices have demonstrated blocking voltages approaching 10 kV, particularly using lateral device architectures. At the current stage of epitaxial growth capability, Ga₂O₃ cannot yet reliably realize thick (>50-80 μm) drift layers with sufficiently low doping (typically <5×10¹⁵ cm⁻³). Therefore, experimental validation of true vertical Ga₂O₃ devices at 10 kV rating remains very limited. In response to the reviewer's suggestion, we preformed electrical simulations of vertical Ga₂O₃ device at 10 kV by scaling drift thickness (50 μm) and doping concentration (2×10¹⁵ cm⁻³), the device edge termination is the same as that in this manuscript. As shown in Fig. R19, at 10 kV, the Ga₂O₃

device presents a peak E -field of 6 MV/cm below BTO field plate, and the main junction region still demonstrated uniform E -field distribution.

Fig. R19. Simulated two-dimensional electric field contours in NiO/Ga₂O₃ heterojunction diode with thick drift layer using the same edge termination.

In the short-pulse regime, increasing the drift-layer thickness does not significantly alter the thermal response, because the generated heat remains locally confined and thermal diffusion is negligible. As a result, the transient temperature rise is primarily governed by the intrinsic volumetric heat capacity of the material. A thicker drift layer introduces a larger thermal capacitance, which can also lower the transient temperature rise. Even under longer pulse or higher frequency conditions, owing to the junction-side cooling configuration, the majority of the heat is extracted through the package rather than through the Ga₂O₃ chip itself. Moreover, the additional drift-layer thickness is relatively insignificant compared with the substrate thickness (650 μm). Therefore, under both short- and long-pulse operations, the module's current handling capability is not expected to be significantly affected, enabling good power scalability of pulsed power capacity for the present module design. The fundamental advantage of Ga₂O₃ module revealed in this work remains effective to higher voltage, further supporting the Ga₂O₃'s promise for ultra-high-power pulsed applications.

Corresponding change in manuscript: No

Comment (3): Following the above point, please also analyze if the power capacity advantage over SiC counterparts retains at higher voltage, as SiC modules have been demonstrated up to 10 kV.

Response: We thank the reviewer for this important follow-up question. Following the discussion above, we expect that the power capacity advantage of Ga₂O₃ modules over their SiC counterparts is retained at higher voltage ratings, including the 10 kV class. First, as clarified in our previous response, the power capacity of the proposed Ga₂O₃ module is not fundamentally limited by the voltage rating. Under pulsed-power operation, the temperature rise is primarily governed by the effective thermal capacitance. Therefore, increasing the blocking voltage does not intrinsically degrade the transient power handling capability of the Ga₂O₃ module.

Second, the fundamental material advantages of Ga₂O₃ over SiC (namely its large volumetric heat capacity and high allowable operating temperature) are largely voltage-rating agnostic. These properties are intrinsic to the material and remain valid for higher-voltage devices, including those rated at 10 kV. Consequently, the superior transient power capacity observed in Ga₂O₃ devices at lower voltages is expected to persist at elevated voltage levels.

Corresponding change in manuscript: No

Comment (4): In Fig. 1f, please clarify the physical mechanism by which the high- κ dielectric introduces dipole polarization charges at both the high- κ /metal and high- κ /Ga₂O₃ interfaces.

Response: We thank the reviewer for the suggestion and have revised the manuscript accordingly. As discussed in **Response (1) of Reviewer #2**, our motivation for selecting BTO is primarily rooted in its high dielectric permittivity and the resulting electric-field modulation functionality under reverse bias. When the Ga₂O₃ drift region becomes depleted, fixed positive charge exists in the depletion region. Owing to the large permittivity contrast between BTO and Ga₂O₃, the external electric field induces a polarization gradient inside the BTO layer, creating net negative bound polarization charge at the BTO/Ga₂O₃ interface and a corresponding positive charge beneath the metal/BTO interface. As shown in Fig. R20, the negative bound charge effectively compensates the positive charge in the depletion region, reducing the local electric-field peak and flattening the field distribution. This permittivity-induced charge balancing is similar to conventional superjunction compensation, where equal positive and negative charges flatten the electric field and decouple breakdown voltage from sheet charge density.

In theory, this polarization superjunction functionality will hold for all high-permittivity dielectrics. Here, BTO is selected not only for its high permittivity (measured to be 155 from our *C-V* characterization, as shown in Supplementary Information S3), but also for its fabrication simplicity, low leakage current, and proven compatibility with Ga₂O₃ to form high-quality heterojunctions. In addition, BTO has been successfully applied in both Ga₂O₃ and GaN based power devices, demonstrating superior capability in improving surface electric-field distribution and enhancing breakdown voltage. Therefore, BTO represents an optimal dielectric choice for this work.

Fig. R20. Illustration of the polarization dipoles introduced by the high- κ dielectrics in the metal/BaTiO₃/Ga₂O₃ structure

Corresponding change in manuscript: Yes

Location of Change: The revisions have been incorporated into the 3rd paragraph of the “**Device-package electrothermal co-optimization**” section (Page 6) in the revised manuscript.

Expanded text:

“As illustrated in **Fig. 1f**, owing to the large permittivity contrast between BTO and Ga₂O₃, the external electric field induces a polarization gradient inside the BTO layer, creating net negative bound polarization charge at the BTO/Ga₂O₃ interface and a corresponding positive charge beneath the metal/BTO interface^{64,65}. This polarization dipole can shield the electric field produced by charges in the metal and enable a nearly-zero net charge to flatten the crowded electric field – similar to the legacy superjunction design in multidimensional power devices⁶⁶.”

Comment (5): In Fig. 3b, the peak temperature appears at the anode edge for submodule B, but at the device center for submodule C. Please explain the difference.

Response: We thank the reviewer for this important observation. The difference in peak-temperature locations between sub-modules B and C arises from their distinct heat-spreading geometries and contact footprints with the cooling structures. In sub-module B, a metal post is inserted on the top surface of the Ga₂O₃ chip. To prevent silver overflow during bonding, the post area is intentionally made smaller than the actual anode metallization area. During pulse conduction, the majority of heat is generated beneath the full anode area, but only the region directly underneath the post provides efficient heat extraction. The peripheral portion of the anode, which is not covered by the post, must dissipate heat only through the Ga₂O₃ layer, resulting in slower cooling at the anode edge and the formation of a localized peak temperature.

In contrast, in sub-module C, the high- κ interface enables continuous silver bonding over the full anode footprint. As a result, heat is extracted more uniformly from both the central and peripheral regions, eliminating edge heating and yielding a much flatter junction temperature distribution. Quantitatively, sub-module B exhibits a local peak of ~89 °C near the edge, whereas submodule C maintains a nearly uniform junction temperature of ~57 °C. This confirms that the improved coverage and thermal spreading path in sub-module C effectively suppress the edge-localized hot spot observed in sub-module B.

Corresponding change in manuscript: Yes

Location of Change: The revisions have been incorporated into the 2nd paragraph of the “**Transient thermal impedance and maximum power capacity**” section (Page 7) in the revised manuscript.

Revised text:

“These hot spots arise because the post contact area is smaller than the actual anode area, resulting in ineffective heat extraction at the anode edge regions.”

Comment (6): Oscillations are observed in both the voltage and current waveforms in Fig. 4f and Fig. 4g following the switching event. Please explain the origin of these oscillations. Additionally, are there any methods that can be applied to mitigate this phenomenon?

Response: We greatly appreciate the reviewer’s insightful comment regarding the waveform oscillations. These oscillations originate from the high-frequency resonance between parasitic inductances and capacitances within the commutation loop during high-speed switching. As illustrated in the equivalent circuit schematic in Fig. R21, the high-current path includes stray inductances from the PCB traces (L_{PCB} , typically 1-5 nH) and the packaging inductances of the terminals (L_D, L_S, L_A, L_C , typically 1-10 nH). When coupled with the parasitic output capacitance of the power switches (C_{DS}) and the junction capacitance of the Ga₂O₃ DUT (C_{DUT}), these components form an RLC resonant tank. During the rapid switching transients required for short-pulse generation, the high di/dt and dv/dt excite this resonant network, resulting in the observed voltage and current ringing.

To mitigate this phenomenon, two primary strategies can be employed. The first is to minimize loop inductance through compact PCB layout and advanced packaging. In our design, we have implemented a minimum-loop PCB layout and utilized our low-inductance Ga₂O₃ module packaging; however, the necessary inclusion of high-voltage, high-current commercial components (e.g., TO-247 packaged SiC MOSFETs) imposes a physical limit on the minimum achievable inductance. The second method is to dampen the oscillation by slowing down the switching speed (e.g., increasing gate resistance) to reduce di/dt. However, doing so would significantly widen the minimum pulse width beyond 5 μ s. Since the primary objective of this work is to characterize the transient thermal superiority of Ga₂O₃ under ultra-short pulse conditions, we prioritized fast switching speeds. Therefore, the waveforms presented in Fig. 4f and Fig. 4g in the main text represent an optimized trade-off between achieving minimal pulse widths and managing parasitic oscillations, with the observed ringing being characteristic of high-speed, megawatt-class dynamic testing.

Fig. R21. Equivalent circuit schematic of switching legs in on-board pulsed power switching test.

Corresponding change in manuscript: No

Comment (7): For the pulse power application discussed, what is the rationale for selecting a pulse width of 3.6 μs ? Furthermore, is it feasible to reduce the pulse width?

Response: We greatly appreciate the reviewer's insightful comment regarding the selection of the pulse width. As detailed in the circuit operation principle (Supplementary Figure S11), the duration of the current pulse applied to the DUT is determined by the off-time of switch S1 during Stage II. The commercial SiC MOSFETs used as control switches exhibit finite rise and fall times; to ensure a well-defined current pulse profile rather than a transient spike, the pulse width must be sufficiently longer than these switching transition periods. Furthermore, as discussed in the previous **Response 6**, high-speed switching induces high-frequency parasitic oscillations. Reducing the pulse width further risks merging the turn-on and turn-off transients, where the resulting ringing could induce false triggering or electromagnetic interference, compromising the safety of the setup. Consequently, the selected pulse width represents an optimized operating point that balances the requirements for short-pulse thermal characterization with the stability and safety of the high-power test platform developed in this work. It is important to note that further reduction of the pulse width is technically feasible by optimizing the test bed with lower-inductance PCB layouts or faster driving stages; however, the limit of the current experimental setup is imposed by the test circuit, not the Ga_2O_3 module itself, which has demonstrated a fast reverse recovery time of 23 ns.

Corresponding change in manuscript: No

6. Reply to the 6th reviewer's comments

Reviewer #6 (Remarks to the Author):

+ What are the noteworthy results?

- This manuscript presents a novel power module based on Ga₂O₃ power devices for pulsed power applications. The module under investigation promises enhanced current/voltage capability as well as careful temperature management. The Authors also assembled a multi-die module allowing switching at 1kA/1kV.

+ Will the work be of significance to the field and related fields? How does it compare to the established literature? If the work is not original, please provide relevant references

- This work is significant to the field, as the proposed sample shows unprecedented features.

+ Does the work support the conclusions and claims, or is additional evidence needed?

- No additional evidence is needed.

+ Are there any flaws in the data analysis, interpretation and conclusions? Do these prohibit publication or require revision?

+ Is the methodology sound? Does the work meet the expected standards in your field?

- The methodologies are sound and meet the standard.

+ Is there enough detail provided in the methods for the work to be reproduced?

- The article as well as the supplementary material provide enough detail for the work to be reproduced.

Response: We sincerely thank the reviewer for the positive evaluation and encouraging remarks. We greatly appreciate the recognition of the novelty, experimental completeness, and methodological rigor of our work. We are pleased that the reviewer finds the results significant, reproducible, and well supported by the presented evidence. Thank you for your constructive and supportive assessment.

- Some minor remarks follow:

Comment (1): In fig. 1b there is a typo: "shot" is written but "short" was intended;

Response: Thank you for pointing out the typographical error. We have corrected "shot" to "short" in Fig. 1b.

Comment (2): Initially, the Authors claim that the value of the thermal conductivity of the assembly materials is not relevant in the case of pulsed power electronics applications, whereas the heat capacity is the most important parameter; however, in the "Device-package electrothermal co-optimization" paragraph, they underline the low-kt value of Ga₂O₃. Fig. 5h also shows that kt plays a relevant role in determining the thermal behavior of the modules. More details are needed to evaluate the individual impact of the two above parameters on the FOMs of the proposed module.

Response: We thank the reviewer for raising this important point. We clarify that thermal conductivity (k_t) and volumetric heat capacity (C_v) both play important roles, but their influence depends on the characteristic time scale of the pulsed power event and the repetition rate (or frequency). For very short pulses with small duty cycles (the usual application profile in pulsed power applications), the temperature rise can be

approximated by: $\Delta T = Q/(C_V V)$ indicating that materials with larger heat capacity experience a smaller instantaneous temperature increase for the same injected energy^{36,37}. As the pulse width increases or the frequency increases, lateral and vertical heat spreading becomes dominant, and the time-dependent transient thermal resistance is strongly influenced by the thermal diffusion length: $L_d = 2\sqrt{(k_t/C_V)t}$, where t is the pulse on-state time^{38,39}. When t becomes larger and diffusion length expands to package case, then heat extraction and k_t become more important. In conventional power switching applications, the pulse width is usually relatively long with a duty cycle of 20~80%, making the material's k_T important for module's thermal performance. In pulsed power applications investigated in this work, the pulse width is usually short (down to a few micro-seconds or below) with a duty cycle usually below 1%, making the material's C_V more important for the module's thermal performance.

To experimentally validate this, we designed the surge current tests with varying pulse widths across several orders of magnitude, and the results are presented in Fig. 5. As shown in Fig. 5h, for pulse widths below 10 μs , the Ga_2O_3 module exhibits lower peak T_j compared to the SiC module benefiting its higher C_V and higher temperature endurance compared to SiC. For pulse widths above 100 μs , SiC module shows lower T_j than Ga_2O_3 module due to SiC's higher k_T . For these long pulse width and steady-state switching applications, a widely-used package design to address the low k_T of Ga_2O_3 (also adopted in this work) is the junction-side-cooling package, which allows heat extraction directly from the device junction instead of through the low- k_T chip.

In this work, despite the major focus being on short pulse width applications where C_V is more relevant, we still adopt the junction-side cooling package due to two reasons. First, it favors efficient heat extraction during the cooling stage in each switching cycle. Second, the junction-side-cooling package can usually allow for a higher thermal capacitance (due to parallel connection of the chip and junction-side package in the equivalent thermal network), which also favors for suppressing the temperature rise in the pulsed-on transient.

Corresponding change in manuscript: No

References:

- [36] Incropera, F. P. & DeWitt, D. P. *Fundamentals of heat and mass transfer*. (6th Edition, J. Wiley & Sons, New York, 2007).
- [37] Heat and Mass Transfer - Fundamentals and Applications, 6th Edition, McGraw-Hill Education, New York, NY, 2020.
- [38] Marín, E. "Characteristic dimensions for heat transfer." *Latin-American Journal of Physics Education* 4.1 (2010): 56-60.
- [39] R. B. Bird, W. E. Stewart, E. N. Lightfoot. *Transport Phenomena*, John Wiley and Sons: New York, 1976.